# Potential clinical relevance of cardiac magnetic resonance to diagnose cardiac light chain amyloidosis

Zsofia Dohy[1], Liliana Szabo[1], Zoltan Pozsonyi[2], Ibolya Csecs[1], Attila Toth[1], Ferenc Imre Suhai[1], Csilla Czimbalmos[1], Andrea Szucs[1], Anna Reka Kiss[1], David Becker[1], Bela Merkely[1☯], Hajnalka Vago[1☯]*

**1** Heart and Vascular Center, Semmelweis University, Budapest, Hungary, **2** Department of Internal Medicine and Haematology, Semmelweis University, Budapest, Hungary

☯ These authors contributed equally to this work.
* vagoha@gmail.com

## Abstract

### Background

While patients with cardiac transthyretin amyloidosis are easily diagnosed with bone scintigraphy, the detection of cardiac light chain (AL) amyloidosis is challenging. Cardiac magnetic resonance (CMR) analyses play an essential role in the differential diagnosis of cardiomyopathies; however, limited data are available from cardiac AL-Amyloidosis. Hence, the purpose of the present study was to analyze the potential role of CMR in the detection of cardiac AL-amyloidosis.

### Methods

We included 35 patients with proved cardiac AL-amyloidosis and two control groups constituted by 330 patients with hypertrophic cardiomyopathy (HCM) and 70 patients with arterial hypertension (HT), who underwent CMR examination. The phenotype and degree of left ventricular (LV) hypertrophy and the amount and pattern of late gadolinium enhancement (LGE) were evaluated. In addition, global and regional LV strain parameters were also analyzed using feature-tracking techniques. Sensitivity and specificity of several CMR parameters were analyzed in diagnosing cardiac AL-amyloidosis.

### Results

The sensitivity and specificity of diffuse septal subendocardial LGE in diagnosing cardiac AL-amyloidosis was 88% and 100%, respectively. Likewise, the sensitivity and specificity of septal myocardial nulling prior to blood pool was 71% and 100%, respectively. In addition, a LV end-diastolic septal wall thickness ≥ 15 mm had an optimal diagnostic performance to differentiate cardiac AL-amyloidosis from HT (sensitivity 91%, specificity 89%). On the other hand, a reduced global LV longitudinal strain (< 15%) plus apical sparing (apex-to-base longitudinal strain > 2) had a very low sensitivity (6%) in detecting AL-Amyloidosis, but with very high specificity (100%).

**Data Availability Statement:** All relevant data are within the paper and its Supporting Information files.

**Funding:** Project no. NVKP_16-1–2016-0017 ('National Heart Program') has been implemented with the support provided from the National Research, Development and Innovation Fund of Hungary, financed under the NVKP_16 funding scheme. The research was financed by the Thematic Excellence Programme (2020-4.1.1.-TKP2020) of the Ministry for Innovation and Technology in Hungary, within the framework of the Therapeutic Development and Bioimaging thematic programmes of the Semmelweis University; and by the Ministry of Innovation and Technology NRDI Office within the framework of the Artificial Intelligence National Laboratory Program. LS was supported by the ÚNKP-20-3-II-SE-61 New National Excellence Program of the Ministry for Innovation and Technology from the source of the National Research, Development and Innovation Fund. ZD and LS were supported by the „Development of scientific workshops of medical, health sciences and pharmaceutical educations" project. Project identification number: EFOP-3.6.3-VEKOP-16-2017-00009. The funders had no role in study design, data collection and analysis, decision to publish, or preparation of the manuscript.

**Competing interests:** The authors declare that no competing interest exist.

## Conclusions

The findings from this study suggest that CMR could have an optimal diagnostic performance in the diagnosis of cardiac AL-amyloidosis. Hence, further larger studies are warranted to validate the findings from this study.

## Introduction

Cardiac involvement of light chain (AL) amyloidosis is characterized by impaired cardiac function, left ventricular (LV) hypertrophy and tissue specific changes of the myocardium. In a background of increased LV wall thickness or LV hypertrophy, several primary and secondary causes can be detected besides cardiac AL-amyloidosis, such as hypertrophic cardiomyopathy (HCM), endomyocardial fibrosis, cardiac involvement of Fabry disease, and pressure overload of the LV [1, 2]. Since the treatment and prognosis of these diseases vary significantly, differential diagnosis is crucial. While patients with transthyretin amyloidosis can be diagnosed with bone scintigraphy, the detection of cardiac involvement in light chain amyloidosis is challenging. Cardiac magnetic resonance (CMR) examinations have an essential role in the diagnosis of myocardial diseases. CMR imaging allows the extraction of morphologic features and the pattern of late gadolinium enhancement (LGE), which are traditionally used to establish the diagnosis of various pathological processes [3–5]. However, in patients contraindicated for contrast agent administration, the diagnosis can be challenging. Novel CMR techniques, including parametric mapping or strain analysis, are available and can help in the differential diagnosis of these patients.

Strain analysis is a useful and reliable method for assessing global and regional myocardial function. Myocardial strain abnormalities assessed by echocardiography have been described and widely accepted to occur in myocardial diseases with LV hypertrophy [6–8]. However, echocardiography-based strain analysis might be challenging for patients with poor acoustic windows, especially when imaging the LV apex. In these cases, CMR imaging can be a useful alternative for imaging the entire LV myocardium. The feature-tracking technique has been validated for strain analysis using standard cine CMR images [9–11]. Previous studies investigated the feature-tracking strain characteristics of cardiomyopathies, the prognostic significance of strain parameters, and the association between LGE and myocardial deformation in different ischemic and nonischemic myocardial diseases [12–15]. However, limited data are available on how feature-tracking strain analysis can help in the differential diagnosis of myocardial diseases causing LV hypertrophy.

Despite the advantages of CMR imaging in the diagnosis of cardiac AL-amyloidosis, there is a lack of comprehensive studies with large study populations that have investigated the role of CMR-based strain analysis in this patient population. Therefore, we conducted a study with the aim of investigating the importance of CMR parameters including feature-tracking strain analysis in differentiating cardiac AL-amyloidosis from HCM and cardiac AL-amyloidosis from myocardial consequences of arterial hypertension.

## Materials and methods

### Patients

We retrospectively identified all patients with myocardial disease causing LV hypertrophy or increased LV wall thickness who were referred to The Heart and Vascular Center of

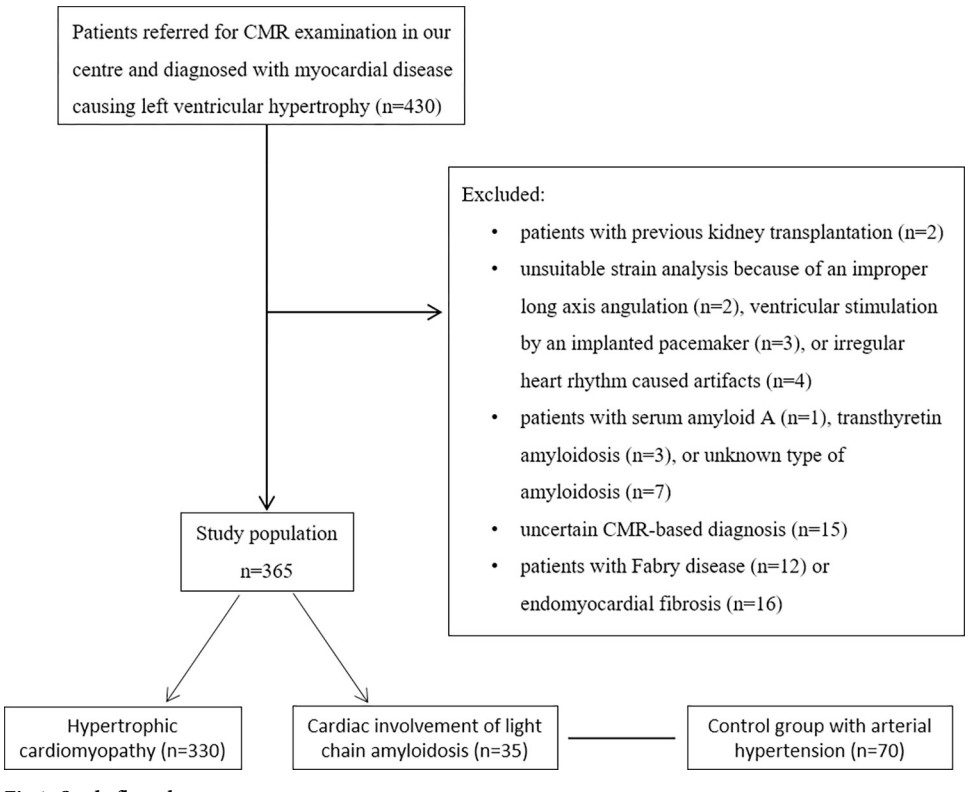

**Fig 1. Study flow chart.**

Semmelweis University between 2009 and 2019 for CMR examination. Patients with significant aortic stenosis, or athletes with left ventricular hypertrophy due to physiological sport adaptation were not involved in the study. The indications for CMR imaging were the assessment of LV hypertrophy identified with other imaging modalities, the detection of cardiac involvement from a known systemic amyloidosis, or the presence of electrocardiographic abnormalities. Patients with HCM or cardiac AL-amyloidosis were involved in the study. Patients with serum amyloid A (n = 1) and transthyretin amyloidosis (n = 3), or if the exact type of amyloidosis was unknown (n = 7) were excluded from the study. Additionally, patients with Fabry disease (n = 12), endomyocardial fibrosis (n = 16), previous kidney transplantation (n = 2), unsuitable strain analysis (n = 9) or an uncertain CMR-based diagnosis (n = 15) were excluded from the study (Fig 1).

As a control group, we selected from our database 70 patients with (treated or untreated) arterial hypertension (HT) without history of cardiomyopathy and with similar LV ejection fraction than the group with cardiac AL-amyloidosis.

Informed consent was obtained from each patient. Ethical approval was obtained from the Hungarian National Institute of Pharmacy and Nutrition (OGYEI/29174-4/2019), and this study was performed in accordance with the ethical standards in the 1964 Declaration of Helsinki and its later amendments.

## CMR protocol

CMR examinations were conducted with a 1.5 T MR scanner (Achieva, Philips Medical Systems and Magnetom Aera, Siemens Healthcare) using a 5-channel cardiac coil. Retrospectively

gated balanced steady-state free precession (bSSFP) cine images were acquired in 2-chamber, 4-chamber and LV outflow tract views. Additionally, short-axis (SA) images with full coverage of the LV were obtained. If no contraindications for contrast agent administration were present, a bolus of gadobutrol (0.15 mmol/kg) was injected at a rate of 2–3 ml/s through an antecubital intravenous line. LGE images were acquired using a segmented inversion recovery sequence with additional phase-sensitive reconstructions in the same views used for the cine images 10–20 minutes after contrast administration.

## Image analysis

CMR data were analyzed using Medis Suite 3.1 software (Medis Medical Imaging Software, Leiden, The Netherlands). The left ventricular ejection fraction (LVEF), volumes (end-diastolic volume: LVEDV, end-systolic volume: LVESV, stroke volume: LVSV), and mass (LVM) were quantified. The LV volumes and LVM were standardized to the body surface area (BSA), yielding LVEDVi, LVESVi, LVSVi, and LVMi. End-diastolic wall thickness (EDWT) measurements were taken in an SA slice perpendicular to the myocardial centerline, excluding trabeculation. The amount of LGE was quantified at a grayscale threshold of 5 standard deviations (SDs) above the mean signal intensity for normal myocardium. LV strain analysis was performed with the feature-tracking application of the MedisSuite: QStrain module. Endocardial contour detection was manually performed on the three long-axis (LA) and SA cine images on basal, midventricular and apical slices during the end-systolic and end-diastolic phases. Global longitudinal (GLS), circumferential (GCS) and radial (GRS) LV strain parameters were measured. Strain values for the six basal, six midventricular, and five apical segments were averaged to obtain regional longitudinal and circumferential strain values (basal LS, midventricular LS, apical LS, basal CS, midventricular CS, apical CS) (Fig 2). The apex-to-base regional LS and CS ratios were calculated as apical LS/basal LS and apical CS/basal CS, respectively. To assess global dyssynchrony, mechanical dispersion (MD) was measured, which was defined as the standard deviation (SD) of the time-to-peak circumferential (MDC) and longitudinal (MDL) strains of the LV segments expressed as percentages of the cardiac cycle. The SDs of the segmental peak LS and CS (SD-LS-Peak and SD-CS-Peak, respectively) were also assessed. Interobserver variability in strain parameters was measured in a subgroup of randomly selected patients (n = 50). Stain parameters with an intraclass correlation higher than 0.6 were accepted for analysis; therefore, SD-CS-Peak and strain parameters concerning myocardial rotation were excluded (S1 Table).

## CMR diagnosis

The CMR diagnosis was made based on the extracted morphologic features and LGE pattern (Fig 3) and was compared to the patient's history. The diagnosis of HCM was based on the finding of a maximal wall thickness ≥ 15 mm in any myocardial segment or a ratio of maximal apical to posterior wall thickness ≥ 1.5 in case of hypertrophy predominating in the LV apex, if no other reason was found causing LV hypertrophy. In the case of a family history of HCM, in first-degree relatives, the diagnosis of HCM was based on the presence of otherwise unexplained increased wall thickness ≥13 mm [1, 16]. The diagnosis of cardiac AL-amyloidosis was confirmed by biopsy and CMR features consistent with cardiac involvement as follows: LV wall thickness >12 mm; diffuse LGE; abnormal gadolinium kinetics typical for cardiac AL-amyloidosis [17, 18]. The diagnosis of FD was proven with enzyme and/or genetic testing. The CMR features of cardiac involvement of FD included LV hypertrophy with or without a typical pattern of LGE in the basal inferolateral segment with midmyocardial distribution [19]. In the case of EMF, LGE was observed in the endocardium mainly in the apex and eventually in the

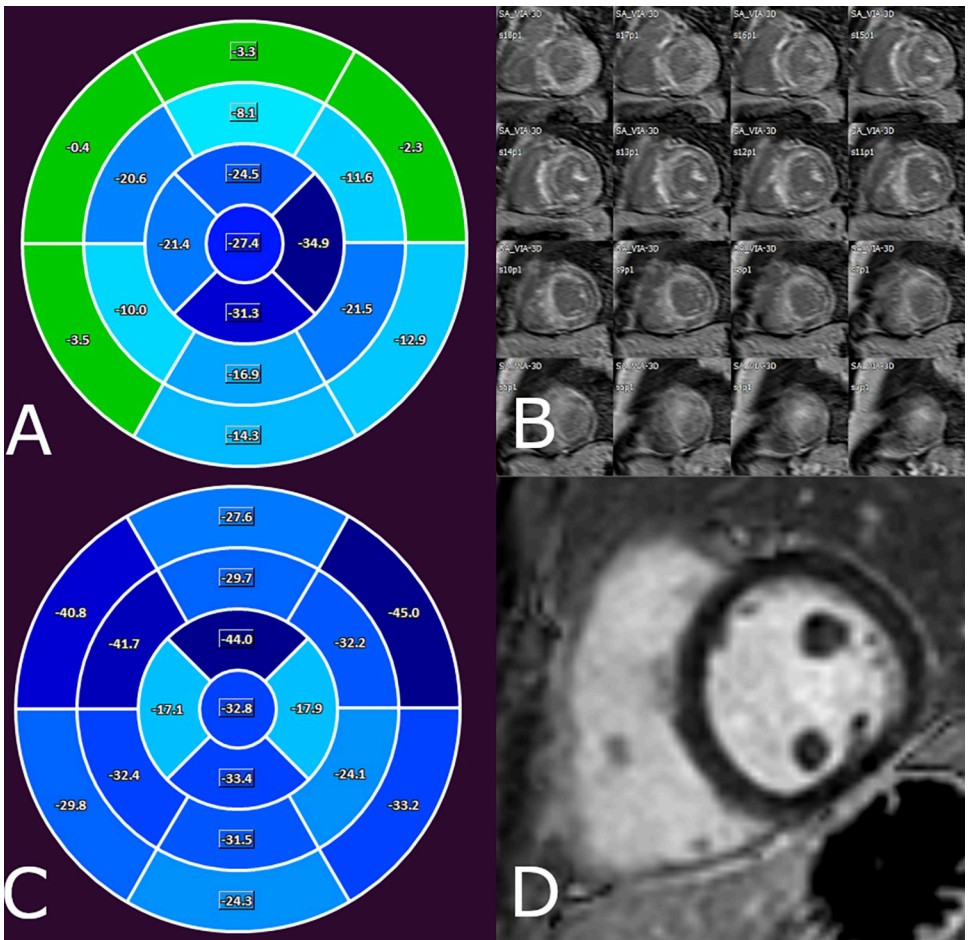

**Fig 2.** Bull's eye with segmental LS values (A, C) and late enhancement images in short-axis slices (B, D) of a patient with cardiac amyloidosis (A, B) and of a person without structural heart diseases (C, D).

subvalvular region of the LV [20]. For all patients, the CMR diagnosis was approved by one of two consultants with >10 years of experience in performing CMR with a European Association of Cardiovascular Imaging CMR level 3 certification.

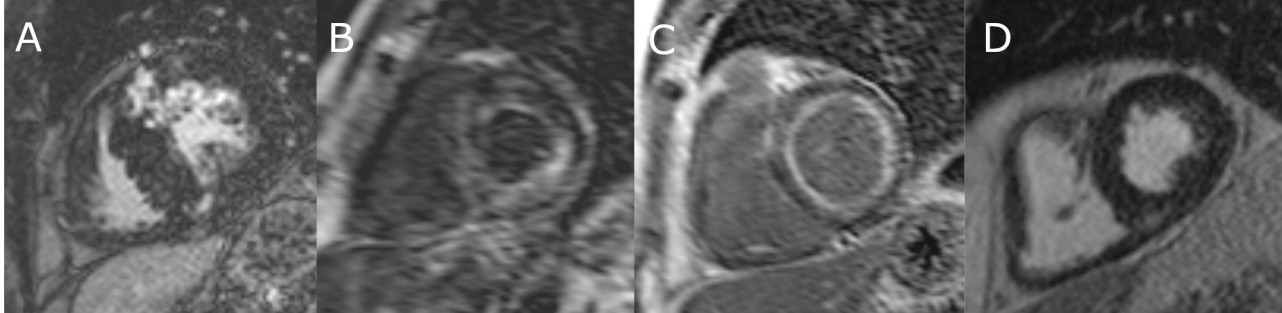

**Fig 3.** Representative late gadolinium enhancement images of patients with hypertrophic cardiomyopathy (A), cardiac AL-amyloidosis (B, C) and arterial hypertension (D) in a short-axis slice. A) Patchy mid-myocardial LGE in the hypertrophic segments typical for HCM (no diffuse subendocardial LGE, normal contrast kinetics). Myocardial nulling prior to blood pool nulling (B), and diffuse subendocardial LGE (C) typical for CA. D) Concentric hypertrophy without diffuse subendocardial LGE and with normal contrast kinetics in a patient with arterial hypertension.

## Statistical analysis

Continuous data are expressed as the mean ± SD. The normality of the distribution of the data was investigated with the Shapiro-Wilk test. Group characteristics were compared with an independent t-test or Mann-Whitney test, as appropriate; and with a Chi-squared test for nominal values. Receiver operating characteristic (ROC) curve analysis was performed to analyze the diagnostic accuracy of a parameter and to identify optimal cut-off values. Differences were considered statistically significant when $p < 0.05$. All analyses were performed by using MedCalc software (version 17.9.5).

## Results

### Patient population

Over a 10-year period, 330 patients were diagnosed with HCM, and 46 patients were diagnosed with cardiac AL-amyloidosis. The most common form of HCM was asymmetric hypertrophy with a septal or an anterior distribution, which was found in 257 patients (77.9%). There were 47 (14.2%) patients with apical HCM, 21 (6.4%) patients with concentric HCM and five (1.5%) patients with midventricular HCM. Among cardiac AL-amyloidosis patients, concentric hypertrophy was found in 16 cases (35%), and hypertrophy showed septal dominance in 30 patients (65%). CMR imaging provided a different diagnosis from the referral diagnosis for 8% of HCM and 26% of cardiac AL-amyloidosis patients.

A control group of 70 patients with HT (treated or untreated) were selected who had similar age, sex rate and LVEF than the cardiac AL-amyloidosis patient group.

### Conventional CMR parameters and feature-tracking strain analysis

The demographic and CMR data of the patient groups are summarized in Table 1. Cardiac AL-amyloidosis patients were older had a lower LVEF and LVSVi, a higher LVESVi, and higher amounts of LGE than HCM patients. There was no difference in LVMi between HCM and cardiac AL-amyloidosis patients; however, HCM patients had higher EDWT. Concentric hypertrophy was more frequent among cardiac AL-amyloidosis patients. Cardiac AL-amyloidosis patients had lower GRS and more impaired global and regional LS and CS values. The apex-to-base CS and LS values were higher in cardiac AL-amyloidosis patients than in HCM patients. There were no differences in the MDC and MDL parameters between cardiac AL-amyloidosis and HCM patients. We found higher GRS/EF ratio in HCM patients.

When comparing cardiac AL-amyloidosis and control group with HT, we found more pronounced LV hypertrophy and higher amount of LGE in patients with cardiac AL-amyloidosis. Cardiac AL-amyloidosis patients had impaired global and regional LS values, while CS parameters were in absolute value higher in this patient group than in controls with HT. The apex-to-base CS and LS values were higher in cardiac AL-amyloidosis patients.

### Diagnostic value of CMR parameters

In the differentiation of cardiac AL-amyloidosis and HCM, the pattern and amount of LGE, the abnormal contrast kinetics had the highest diagnostic accuracies, followed by basal CS, basal LS, and GRS. The sensitivity and specificity of CMR parameters to differentiate cardiac AL-amyloidosis from HCM are shown in Table 2. The presence of septal or septal and posterior diffuse subendocardial LGE had high specificity (99% for both) and relatively high sensitivity (88% for both). The specificity of myocardial nulling prior to blood pool nulling or difficulty in achieving myocardial nulling was 100%, with a sensitivity of 71%. The results of ROC analyses are shown in S2 Table. The optimal cut-off values of the above mentioned

**Table 1. Demographic and CMR characteristics of the study population.**

| | HCM (n = 330) | Cardiac AL-amyloidosis (n = 35) | HT (n = 70) | p | p |
|---|---|---|---|---|---|
| | mean±SD | mean±SD | mean±SD | Cardiac AL-amyloidosis vs. HCM | Cardiac AL-amyloidosis vs. HT |
| age | 46.6±18.3 | 64.1±9.2 | 59.7±12.1 | <0.0001 | 0.06 |
| sex (male%) | 61.5 | 64.3 | 50 | 0.41 | 0.68 |
| BSA (m$^2$) | 1.94±0.29 | 1.86±0.24 | 1.99±0.30 | 0.11 | 0.054 |
| LVEF (%) | 63.6±7.3 | 51.0±11 | 54.7±8.6 | <0.0001 | 0.06 |
| LVEDVi (ml/m$^2$) | 86.9±17.3 | 82.6±18.8 | 86.7±23 | 0.13 | 0.5 |
| LVESVi (ml/m$^2$) | 31.9±10.2 | 41±15.2 | 40.5±17.7 | <0.001 | 0.45 |
| LVSVi (ml/m$^2$) | 55.1±11.3 | 41.6±11.8 | 46.2±9.4 | <0.0001 | <0.05 |
| LVMi (g/m$^2$) | 89.2±32.9 | 88.3±18.3 | 54.3±15.8 | 0.5 | <0.0001 |
| max. EDWT (mm) | 20.2±4.9 | 17.3±2.2 | 11.5±2.2 | <0.001 | <0.0001 |
| LGE% | 8.3±8.4 | 27.1±14.8 | 0.9±1.8 | <0.0001 | <0.0001 |
| GRS (%) | 87.2±24.7 | 55.1±22.3 | 57.6±17.8 | <0.0001 | 0.53 |
| GCS (%) | -40.9±8.7 | -32.9±10.1 | -27.7±6.5 | <0.0001 | <0.01 |
| GLS (%) | -23.7±5.7 | -18.4±4.6 | -21.6±4.2 | <0.0001 | <0.001 |
| SD-LS-Peak | 12.2±2.7 | 10.6±2.8 | 11.1±5.7 | <0.01 | 0.47 |
| MDC (%) | 6.9±3.8 | 6.9±3.2 | 9.2±4.7 | 0.7 | <0.05 |
| MDL (%) | 16.2±5.4 | 17.1±5 | 11.8±4.2 | 0.34 | <0.0001 |
| basal CS (%) | -37.7±7.2 | -26.1±8.7 | -27±6 | <0.0001 | 0.87 |
| mid CS (%) | -38.9±9.1 | -29.7±9.8 | -25.2±6.6 | <0.0001 | <0.05 |
| apical CS (%) | -47.3±12.9 | -41.7±14.2 | -30.8±8.8 | <0.05 | <0.0001 |
| apex-to-base CS | 1.28±0.37 | 1.61±0.64 | 1.15±0.26 | <0.001 | <0.0001 |
| basal LS (%) | -21.3±5.9 | -15±3.7 | -25.2±5.5 | <0.0001 | <0.0001 |
| mid LS (%) | -24.6±8.9 | -20.1±6.1 | -26.6±5.7 | <0.01 | <0.0001 |
| apical LS (%) | -30.1±8.9 | -25.3±7.4 | -24±7.2 | <0.001 | 0.39 |
| apex-to-base LS | 1.53±0.67 | 1.77±0.61 | 1.00±0.39 | <0.05 | <0.0001 |
| GLS/EF | -0.37±0.09 | -0.36±0.05 | -0.39±0.04 | 0.17 | <0.01 |
| GCS/EF | -0.65±0.1 | -0.64±0.13 | -0.50±0.07 | 0.86 | <0.0001 |
| GRS/EF | 1.36±0.33 | 1.05±0.29 | 1.03±0.22 | <0.0001 | 0.99 |

Comparison of the parameters of patients with different diagnoses with an independent t-test or Mann-Whitney test, as appropriate.

parameters are as follows: LGE% cut-off of 16% (sensitivity: 76%, specificity: 87%, AUC: 0.916), basal CS cut-off of -31% (sensitivity: 71%, specificity: 83%, AUC: 0.874), basal LS cut-off of -16% (sensitivity: 69%, specificity: 85%, AUC: 0.847), GRS cut-off of 74% (sensitivity: 83%, specificity: 70%, AUC: 0.847).

In the differentiation of cardiac AL-amyloidosis and controls with HT, the degree of hypertrophy, the pattern and amount of LGE, the abnormal contrast kinetics, basal LS and the apex-to-base LS ratio had the highest diagnostic accuracies (Table 3 and S2 Table). The specificity of the presence of septal or septal and posterior diffuse subendocardial LGE and of myocardial nulling prior to blood pool nulling or difficulty in achieving myocardial nulling was 100% with a sensitivity of 88% and 71%, respectively. A minimal amount of LGE was present in 25% of controls with HT, LGE% higher than 6% was strongly diagnostic for cardiac AL-amyloidosis (sensitivity: 97%, specificity: 98%, AUC: 0.995). The optimal cut-off values of LV hypertrophy parameters, basal LS and the apex-to-base LS ratio are as follows (see also S2 Table): max. EDWT cut-off of 14 mm (sensitivity: 94%, specificity: 89%, AUC: 0.967), LVMi cut-off of 61 g/m$^2$ (sensitivity: 100%, specificity: 74%, AUC: 0.927), basal LS cut-off of -21% (sensitivity: 94%,

**Table 2.**

| | Cardiac AL-amyloidosis vs. HCM | | |
|---|---|---|---|
| | sensitivity | specificity | AUC |
| Septal and posterior EDWT ≥ 15 mm | 29% | 93% | 0.610 |
| Septal and posterior EDWT ≥ 14 mm | 31% | 88% | 0.595 |
| Septal and posterior EDWT ≥ 13 mm | 37% | 81% | 0.590 |
| Septal and posterior EDWT ≥ 12 mm | 57% | 69% | 0.630 |
| Septal EDWT ≥ 20 mm | 9% | 60% | 0.345 |
| Septal EDWT ≥ 18 mm | 34% | 46% | 0.400 |
| Septal EDWT ≥ 15 mm | 91% | 19% | 0.550 |
| Septal EDWT ≥ 14 mm | 91% | 13% | 0.520 |
| Septal EDWT ≥ 13 mm | 94% | 9% | 0.515 |
| Septal EDWT ≥ 12 mm | 97% | 5% | 0.510 |
| Septal and posterior diffuse subendocardial LGE | 88% | 99% | 0.935 |
| Septal diffuse subendocardial LGE | 88% | 99% | 0.935 |
| Septal and posterior myocardial nulling prior to blood pool nulling or difficulty in achieving myocardial nulling | 71% | 100% | 0.855 |
| Septal myocardial nulling prior to blood pool nulling or difficulty in achieving myocardial nulling | 71% | 100% | 0.855 |
| Apex-to-base LS > 2 | 31% | 80% | 0.555 |
| Apex-to-base LS > 1.45 | 71% | 49% | 0.609 |
| GLS > -15% and apex-to-base LS > 2 | 6% | 100% | 0.530 |
| GLS > -23% and apex-to-base LS > 1.45 | 57% | 84% | 0.705 |
| GLS > -23% | 86% | 63% | 0.803 |
| GLS > -15% | 23% | 93% | 0.580 |
| GLS > -13% | 17% | 96% | 0.565 |
| GLS > -12% | 9% | 96% | 0.525 |

The sensitivity and specificity of CMR parameters to differentiate cardiac AL-amyloidosis from HCM.

specificity: 79%, AUC: 0.939), apex-to-base LS cut-off of 1.17 (sensitivity: 83%, specificity: 76%, AUC: 0.860).

## Discussion

Analyzing a cohort of 35 patients with proved cardiac AL-amyloidosis and two control groups constituted by 330 patients with hypertrophic cardiomyopathy (HCM) and 70 patients with arterial hypertension (HT), who underwent CMR examination, the findings from this study suggest that CMR could have an optimal diagnostic performance in the diagnosis of cardiac AL-amyloidosis. In this respect, the sensitivity and specificity of diffuse septal subendocardial LGE and of septal myocardial nulling prior to blood pool in diagnosing cardiac AL-amyloidosis was excellent. In addition, a LV end-diastolic septal wall thickness ≥ 15 mm had an optimal diagnostic performance to differentiate cardiac AL-amyloidosis from HT. On the other hand, a reduced global LV longitudinal strain (< 15%) plus apical sparing had a very low sensitivity (6%) in detecting AL-Amyloidosis, but with very high specificity (100%).

The diagnosis of cardiac AL-amyloidosis with CMR examination is traditionally based on the LV hypertrophy phenotype and the pattern of LGE [17, 18]. In our study population, it was found that the amount and pattern of LGE had the highest diagnostic accuracy in the differentiation of cardiac AL-amyloidosis from controls with HT or from HCM. Septal and posterior diffuse subendocardial LGE had a sensitivity and specificity of 88% and 99%, respectively, when differentiating cardiac AL-amyloidosis from HCM, and 88% and 100%, respectively,

**Table 3.**

| | Cardiac AL-amyloidosis vs. HT | | |
|---|---|---|---|
| | sensitivity | specificity | AUC |
| Septal and posterior EDWT ≥ 15 mm | 29% | 100% | 0.645 |
| Septal and posterior EDWT ≥ 14 mm | 31% | 100% | 0.655 |
| Septal and posterior EDWT ≥ 13 mm | 37% | 100% | 0.685 |
| Septal and posterior EDWT ≥ 12 mm | 57% | 97% | 0.770 |
| Septal EDWT ≥ 20 mm | 9% | 100% | 0.545 |
| Septal EDWT ≥ 18 mm | 34% | 100% | 0.670 |
| Septal EDWT ≥ 15 mm | 91% | 89% | 0.900 |
| Septal EDWT ≥ 14 mm | 91% | 83% | 0.870 |
| Septal EDWT ≥ 13 mm | 94% | 73% | 0.835 |
| Septal EDWT ≥ 12 mm | 97% | 54% | 0.755 |
| Septal and posterior diffuse subendocardial LGE | 88% | 100% | 0.940 |
| Septal diffuse subendocardial LGE | 88% | 100% | 0.940 |
| Septal and posterior myocardial nulling prior to blood pool nulling or difficulty in achieving myocardial nulling | 71% | 100% | 0.855 |
| Septal myocardial nulling prior to blood pool nulling or difficulty in achieving myocardial nulling | 71% | 100% | 0.855 |
| Apex-to-base LS > 2 | 31% | 99% | 0.650 |
| Apex-to-base LS > 1.17 | 83% | 76% | 0.860 |
| GLS > -15% and apex-to-base LS > 2 | 6% | 100% | 0.530 |
| GLS > -20% and apex-to-base LS > 1.17 | 49% | 96% | 0.721 |
| GLS > -20% | 66% | 64% | 0.691 |
| GLS > -15% | 23% | 93% | 0.580 |
| GLS > -13% | 17% | 97% | 0.570 |
| GLS > -12% | 9% | 99% | 0.540 |

The sensitivity and specificity of CMR parameters to differentiate cardiac AL-amyloidosis from controls with HT.

when differentiating it from controls with HT. A recent meta-analysis based on 18 published studies included 1,108 cardiac amyloidosis patients (69% were AL) and 907 control subjects, estimated a sensitivity and specificity of 84% and 80%, respectively, for LGE in diagnosing cardiac amyloidosis [21]. According to another meta-analysis of 7 studies, the sensitivity and specificity of LGE CMR in diagnosing cardiac amyloidosis were 85% and 92%, respectively [22]. Expert consensus recommendations [17, 18] state that CMR has a central role in the non-invasive diagnosis of cardiac amyloidosis referring to several studies in which typical LGE pattern has been shown to have a diagnostic sensitivity of 85% to 90% [23–27].

However, in case of a contraindication for contrast agent administration, further diagnostic methods are needed. In recent years, novel CMR techniques, such as mapping measurements, have been developed for the quantitative assessment of myocardial changes. Cardiac amyloidosis is characterized by pronouncedly increased native T1 values. In the case of contrast administration, the extracellular volume of the myocardium can be evaluated with T1 mapping. An increase in extracellular volume is an early marker of cardiac amyloidosis even before the appearance of LGE [28, 29]. Unfortunately, mapping measurements were available in our center only in a few cases for the current study.

We found that strain parameters have relatively high diagnostic accuracy. In the differentiation of cardiac AL-amyloidosis from HCM, the sensitivities of GLS (cut-off of -23%) and GRS (cut-off of 63%) were 89% and 83%, respectively, while the specificities of basal LS (cut-off of -16%), basal CS (cut-off of -31%) were 85% and 83%, respectively. In the differentiation of

cardiac AL-amyloidosis from controls with HT, the sensitivity and specificity of basal LS (cut-off of -21%) were 94% and 79%, respectively, the sensitivity and specificity of apex-to-base LS ratio (cut-off of 1.17) were 83% and 76%, respectively.

In the diagnosis of cardiac amyloidosis, echocardiography-based strain analysis is widely accepted. A well-known typical sign of cardiac amyloidosis is apical sparing, in which basal LS is severely impaired while apical LS is relatively spared [17, 18]. However, only a few studies have investigated the CMR-based strain patterns of cardiac amyloidosis, and the results are controversial. Williams et al. indicated that cardiac amyloidosis patients have worse GLS than HCM patients, but they found no difference in the apex-to-base LS ratio between cardiac amyloidosis and HCM patients [30]. In another study, cardiac amyloidosis patients were compared to healthy controls. Cardiac amyloidosis patients had impaired global, basal, midventricular and apical strain values, but no differences were found in the apex-to-base ratios between cardiac amyloidosis patients and controls; furthermore, the LS values were not different between the apical and basal regions [31]. Bhatti et al. investigated multiple myeloma patients with and without cardiac amyloidosis and found that the apex-to-base gradient was suggestive of apical sparing in patients with cardiac amyloidosis compared with those without cardiac amyloidosis, but no differences were found in the CS and RS values [32]. A recently published study investigated the ability of a single heartbeat fast-strain encoded (SENC) CMR-derived myocardial strain to discriminate between cardiac amyloidosis, HCM, hypertensive heart disease, athletes' heart and healthy controls. Cardiac amyloidosis patients had the most impaired GLS and GCS values, and the percentage of LV segments with a strain value < -17% was the lowest in this patient group; apical sparing was not investigated [33].

We found that cardiac AL-amyloidosis patients have more impaired global and regional LS values than controls with HT or HCM patients. Furthermore, our results demonstrate that feature-tracking strain analysis is applicable for detecting apical sparing in cardiac AL-amyloidosis patients, as they had significantly higher apex-to-base LS and CS ratios than controls with HT and HCM patients. However, in the differentiation of cardiac AL-amyloidosis from HCM, the apex-to-base CS and LS ratios were less accurate than the global and basal strain values, while the diagnostic accuracy of apex-to-base LS ratio was relatively high when differentiating cardiac AL-amyloidosis from controls with HT.

## Limitations

The main advantage of feature-tracking strain analysis is that it needs no additional dedicated CMR sequences, and the evaluation is performed using the standard cine images. However, this method has some limitations: previously published data showed that reliability and accuracy of feature-tracking analysis is dependent on reader experience more than tagging-based strain analysis, and the reproducibility of segmental assessment of strain is lower [34–36].

Another limitation of our study includes its single-center setting. Additionally, myocardial T1 and T2 mapping and myocardial extracellular volume measurements were available only in a few cases of the study population. Finally, in the vast majority of HCM patients, no genetic testing was performed.

## Conclusion

The findings from this study suggest that CMR could have an optimal diagnostic performance in the diagnosis of cardiac AL-amyloidosis. In this respect, the sensitivity and specificity of diffuse septal subendocardial LGE in diagnosing cardiac AL-amyloidosis was 88% and 100% and of septal myocardial nulling prior to blood pool was 71% and 100%, respectively. In addition, a LV end-diastolic septal wall thickness ≥ 15 mm had an optimal diagnostic performance to

differentiate cardiac AL-amyloidosis from HT (sensitivity 91%, specificity 89%). On the other hand, a reduced global LV longitudinal strain ($< 15\%$) plus apical sparing (apex-to-base LS $> 2$) had a very low sensitivity (6%) in detecting AL-Amyloidosis, but with very high specificity (100%). Hence, further larger studies are warranted to validate the potential key role of CMR in the diagnosis of cardiac AL-amyloidosis.

## Supporting information

**S1 Table. Reproducibility of stain analyses.** Intraclass correlation analysis for interobserver variability in strain parameters.
(DOCX)

**S2 Table. Diagnostic accuracy of CMR parameters in differentiating cardiac AL-amyloidosis from HCM and cardiac AL-amyloidosis from controls with HT.** Results of the ROC curve analyses.
(DOCX)

**S1 Dataset. Study's minimal data set.**
(XLSX)

## Author Contributions

**Conceptualization:** Hajnalka Vago.

**Formal analysis:** Zsofia Dohy, Liliana Szabo.

**Funding acquisition:** Bela Merkely.

**Investigation:** Zsofia Dohy, Liliana Szabo, Zoltan Pozsonyi, Ibolya Csecs, Attila Toth, Ferenc Imre Suhai, Csilla Czimbalmos, Andrea Szucs, Hajnalka Vago.

**Supervision:** Zoltan Pozsonyi, David Becker, Bela Merkely, Hajnalka Vago.

**Visualization:** Zsofia Dohy.

**Writing – original draft:** Zsofia Dohy.

**Writing – review & editing:** Liliana Szabo, Zoltan Pozsonyi, Ibolya Csecs, Attila Toth, Ferenc Imre Suhai, Csilla Czimbalmos, Andrea Szucs, Anna Reka Kiss, David Becker, Hajnalka Vago.

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
