## [Decision Letter · Decision Letter 0]

8 Oct 2021

PONE-D-21-24457The Role of Cardiac Magnetic Resonance-based Feature-tracking Strain Analysis in the Differential Diagnosis and Prognostic Assessment of Patients with Left Ventricular Hypertrophy.PLOS ONE

Dear Dr. Vago,

Thank you for submitting your manuscript to PLOS ONE. After careful consideration, we feel that it has merit but does not fully meet PLOS ONE’s publication criteria as it currently stands. Therefore, we invite you to submit a revised version of the manuscript that addresses the points raised during the review process.

Please submit your revised manuscript by Nov 22 2021 11:59PM. If you will need more time than this to complete your revisions, please reply to this message or contact the journal office at plosone@plos.org. Please include the following items when submitting your revised manuscript:A rebuttal letter that responds to each point raised by the academic editor and reviewer(s). You should upload this letter as a separate file labeled 'Response to Reviewers'.A marked-up copy of your manuscript that highlights changes made to the original version. You should upload this as a separate file labeled 'Revised Manuscript with Track Changes'.An unmarked version of your revised paper without tracked changes. You should upload this as a separate file labeled 'Manuscript'.

We look forward to receiving your revised manuscript.

Kind regards,

Daniel A. Morris, M.D

Academic Editor

PLOS ONE

Additional Editor Comments:

Thank you very much for submitting this large study to PlosOne. While the cohort and analyses performed are analyzed interesting, several pending serious and major limitations should be addressed in order to get adequate clinical applicability of the findings from this study.

Pending Major Limitations and Comments:

1) Concerns stated by the reviewer:

- The reviewer has addressed important limitations from this study, which should be mandatorily addressed in the revised version.

3) Uncertainty of the clinical applicability of the LV strain parameters:

- As it is shown in table 2, the rate of false positives and false negatives varied from 20 to 40% between the diverse strain parameters, which obligates to research alternative parameters to accurately differentiate HCM from CA. Hence, the authors should further analyze the diagnostic performance of ECV, T1 native mapping time, and LV phenotype to mainly differentiate HCM from CA.

4) Lack of incremental value analyses:

- The authors should show the sensibility, specificity, accuracy of the following parameters or findings to differentiate HCM from CA:

- Mid Septal ECV > 0,40

- Mid Septal ECV > 0,30

- Mid septal T1 native mapping time > 1200 ms

- Mid septal T1 native mapping time > 1000 ms

- Septal and posterior wall thickness ≥ 15mm

- Septal wall thickness ≥ 25mm

- Septal wall thickness ≥ 15mm

- Septal wall thickness ≥ 14mm

- Septal wall thickness ≥ 12mm

- Septal and posterior diffuse subendocardial LGE

- Septal diffuse subendocardial LGE

- Septal and posterior myocardial nulling prior to blood pool nulling or difficulty in achieving myocardial nulling.

- Septal myocardial nulling prior to blood pool nulling or difficulty in achieving myocardial nulling.

- Apical sparing using LV global longitudinal strain (GLS) (i.e., ratio average apical segments to average basal segments > 2).

- GLS < 15% and apical sparing

- Septal and posterior wall thickness ≥ 14mm + GLS < 15% and apical sparing + pericardial effusion

- Septal and posterior wall thickness ≥ 14mm + GLS < 15% and apical sparing + RV free wall thickness ≥ 7mm

- Septal and posterior wall thickness ≥ 12mm + GLS < 15% and apical sparing + RV free wall thickness ≥ 5mm

“Project no. NVKP_16-1–2016-0017 (’National Heart Program’) has been implemented with the support provided from the National Research, Development and Innovation Fund of Hungary, financed under the NVKP_16 funding scheme. The research was financed by the Thematic Excellence Programme (2020-4.1.1.-TKP2020) of the Ministry for Innovation and Technology in Hungary, within the framework of the Therapeutic Development and Bioimaging thematic programmes of the Semmelweis University; and by the Ministry of Innovation and Technology NRDI Office within the framework of the Artificial Intelligence National Laboratory Program. LS was supported by the ÚNKP-20-3-II-SE-61 New National Excellence Program of the Ministry for Innovation and Technology from the source of the National Research, Development and Innovation Fund. ZD and LS were supported by the „Development of scientific workshops of medical, health sciences and pharmaceutical educations” project. Project identification number: EFOP-3.6.3-VEKOP-16-2017-00009.

We note that you have provided additional information within the Funding Section. Please note that funding information should not appear in other areas of your manuscript. We will only publish funding information present in the Funding Statement section of the online submission form.

“Project no. NVKP_16-1–2016-0017 (’National Heart Program’) has been implemented with the support provided from the National Research, Development and Innovation Fund of Hungary, financed under the NVKP_16 funding scheme. The research was financed by the Thematic Excellence Programme (2020-4.1.1.-TKP2020) of the Ministry for Innovation and Technology in Hungary, within the framework of the Therapeutic Development and Bioimaging thematic programmes of the Semmelweis University; and by the Ministry of Innovation and Technology NRDI Office within the framework of the Artificial Intelligence National Laboratory Program. LS was supported by the ÚNKP-20-3-II-SE-61 New National Excellence Program of the Ministry for Innovation and Technology from the source of the National Research, Development and Innovation Fund. ZD and LS were supported by the „Development of scientific workshops of medical, health sciences and pharmaceutical educations” project. Project identification number: EFOP-3.6.3-VEKOP-16-2017-00009. The funders had no role in study design, data collection and analysis, decision to publish, or preparation of the manuscript.”

5) In order to refine and focus the findings, please exclude patients with Fabry disease and endomyocardial fibrosis.

Reviewers' comments:

Reviewer #1: 

In the present paper Dohy and Merkely et al investigate the ability of myocardial strain by feature tracking analysis for the diagnostic classification and risk stratification of patients with myocardial hypertrophy due to HCM, cardiac amyloidosis (AL and ATTR), Fabry disease and endomyocardial fibrosis. Myocardial strain variables were able to differentiate between HCM and cardiac amyloidosis, whereas risk stratification could also be provided. The paper is surely of interest since strain is an important issue with cardiac imaging and the differentiation between diseases causing hypertrophy has nowadays important therapeutic implications for the specific management of such patients more than ever. However, some specific points definitely need to be clarified.

1. The percentage of patients with HCM is strikingly high compared to those with cardiac amyloidosis. Especially ATTR patients are expected to be found much more frequently in a tertiary setting and AL patients in conjunction with hematology / oncology units. Please explain and consider reporting / acknowledging referral biases.

2. In the same direction, the number of ATTR patients is strikingly low with 3 patients in total! This is certainly not in agreement with current trends of diagnosis this entity in 30-fold more patients than in the last decade. The prevalence especially in elderly patients with hypertrophy of unknow origin is very high, if correctly diagnosed based on current algorithms.

3. In the same line the authors did not include patients with hypertensive heart disease, where in many cases differential diagnosis regarding HCM and cardiac amyloidosis may be very challenging, especially in those with progressed hypertensive heart disease, where focal LGE may be present, mimicking HCM. This needs to be explained by the authors, since inclusion of such patients would have been clinically meaningful and since inclusion of patients has performed retrospectively, which means that such patients could have been easily included and considered for feature tracking analysis too.

4. The same restrictions also apply for individuals with athletes’ heart, while in this case referral may be an issue.

5. Figure 1. It is striking that patients with hypertensive heart disease are not included. Please see also my comment #3.

6. Figure 2. It would be meaningful to parallel show the images of a normal volunteer for comparison to this cardiac amyloidosis patient with very progressed myocardial disease.

7. Diagnosis of HCM. What about patients with septal wall thickness >13mm and family history of HCM?

8. I am not sure if inclusion of patients with endomyocardial fibrosis is meaningful since hypertrophy is not necessarily associated with this entity. Possibly some overlap with hypertensive heart disease needs to be considered with these patients more than with other categories.

9. The differential diagnosis of HCM versus cardiac amyloidosis is clinical meaningful and a strength of the article. It is correct to highlight this in the results, as shown in Table 2. Hereby, LGE seems to be the stronger predictor, which can be expected. However, the value of strain is also relevant since basal LS und CS also provide relatively high AUC values.

10. Figure 4 demonstrates that patients with amyloidosis have poorer prognosis, which is expected due to the entity of the disease. However, much more meaningful would be a Kaplan-Maier analysis based on LGE and on strain values. Would one of the myocardial strain values be able to show prognostic relevance. If yes, this would surely contribute to the current literature.

11. In the same direction, basal LS seems to bear independent prognostic implications for the estimation of mortality. This needs to be tested and adjusted for quantitative LGE and main diagnosis. The corresponding Kaplan-Maier analysis needs to be demonstrated.

12. A recent study investigating the ability of strain based CMR (with fast-SENC) to differentiate between patients with HCM, amyloidosis and hypertensive heart disease needs to be reported and discussed by the authors (Giusca et al, JCMR 2021).

13. In addition, some limitations need to be reported with feature tracking acquisitions. Although, a big advantage of feature tracking is no need for additional dedicated CMR sequences, some disadvantages have been extensively mentioned in the recent literature, especially regarding the segmental assessment of strain (Feisst A et al, IJC Heart Vasc. 2018; Mangion K et al, Sci Rep. 2019 and Almitairi HM et al, Br. J. Radiol. 2017).

---

## [Author Response · Author response to Decision Letter 0]

22 Nov 2021

1) Concerns stated by the reviewer:

- The reviewer has addressed important limitations from this study, which should be mandatorily addressed in the revised version.

Thank you for your comment. The limitations section has been completed with the followings:

“The main advantage of feature-tracking strain analysis is that it needs no additional dedicated CMR sequences, and the evaluation is performed using the standard cine images. However, this method has some limitations: previously published data showed that reliability and accuracy of feature-tracking analysis is dependent on reader experience more than tagging-based strain analysis, and the reproducibility of segmental assessment of strain is lower. There might be some referral bias that can explain the relatively low prevalence of CA patients, especially ATTR patients: ATTR patients are usually referred from community cardiac services or from other cardiology centers where diagnostic awareness is lower compared to hematology centers where myeloma and MGUS patients are monitored for CA and from where a great number of AL patients are referred. In recent years, cardiology centers have diagnosed TTR CA in a "non-biopsy" manner in many cases with the help of PYP scans. Many of these TTR CA pts did not have a CMR for the diagnosis.”

3) Uncertainty of the clinical applicability of the LV strain parameters:

- As it is shown in table 2, the rate of false positives and false negatives varied from 20 to 40% between the diverse strain parameters, which obligates to research alternative parameters to accurately differentiate HCM from CA. Hence, the authors should further analyze the diagnostic performance of ECV, T1 native mapping time, and LV phenotype to mainly differentiate HCM from CA.

Thank you for your comment. We agree with the Editor that ECV and T1 mapping parameters increase the diagnostic accuracy of CMR in the differentiation of CA from HCM. Unfortunately, mapping measurements were not available in our center at the time of the investigated examinations. It has been signed in the limitation section. 

The discussion section has been completed with the followings: “In recent years, novel CMR techniques, such as mapping measurements, have been developed for the quantitative assessment of myocardial changes. CA is characterized by pronouncedly increased native T1 values. In the case of contrast administration, the extracellular volume of the myocardium can be evaluated with T1 mapping. An increase in extracellular volume is an early marker of CA even before the appearance of LGE. Unfortunately, mapping measurements were not available in our center for the current study.”

Regarding LV phenotype, we have completed the results section with the followings: 

“The most common form of HCM was asymmetric hypertrophy with a septal or an anterior distribution, which was found in 257 patients (77.9%). There were 47 (14.2%) patients with apical HCM, 21 (6.4%) patients with concentric HCM and five (1.5%) patients with midventricular HCM. Among CA patients, concentric hypertrophy was found in 16 cases (35%), and hypertrophy showed septal dominance in 30 patients (65%).”

“There was no difference in LVMi between HCM and CA patients; however, HCM patients had higher EDWT. Concentric hypertrophy was more frequent among CA patients.”

4) Lack of incremental value analyses:

- The authors should show the sensibility, specificity, accuracy of the following parameters or findings to differentiate HCM from CA:

- Mid Septal ECV > 0,40

- Mid Septal ECV > 0,30

- Mid septal T1 native mapping time > 1200 ms

- Mid septal T1 native mapping time > 1000 ms

- Septal and posterior wall thickness ≥ 15mm

- Septal wall thickness ≥ 25mm

- Septal wall thickness ≥ 15mm

- Septal wall thickness ≥ 14mm

- Septal wall thickness ≥ 12mm

- Septal and posterior diffuse subendocardial LGE

- Septal diffuse subendocardial LGE

- Septal and posterior myocardial nulling prior to blood pool nulling or difficulty in achieving myocardial nulling.

- Septal myocardial nulling prior to blood pool nulling or difficulty in achieving myocardial nulling.

- Apical sparing using LV global longitudinal strain (GLS) (i.e., ratio average apical segments to average basal segments > 2).

- GLS < 15% and apical sparing

- Septal and posterior wall thickness ≥ 14mm + GLS < 15% and apical sparing + pericardial effusion

- Septal and posterior wall thickness ≥ 14mm + GLS < 15% and apical sparing + RV free wall thickness ≥ 7mm

- Septal and posterior wall thickness ≥ 12mm + GLS < 15% and apical sparing + RV free wall thickness ≥ 5mm

Thank you for your advice. We added the required parameters to a new table (except mapping values) and completed the results section as follows. However, the results of the performed ROC curve analyses showed other cut-off values for GLS (>-23) and apex-to-base LS ratio (>1.36), as recommended by the Editor. Therefor we presented data with our cut-off values as well.

“The frequencies of positive results of different diagnostic criteria in the patient groups are shown in Table 3. The presence of septal or septal and posterior diffuse subendocardial LGE had high specificity (99% and 99.4%, respectively) and relatively high sensitivity (89% for both). The specificity of myocardial nulling prior to blood pool nulling or difficulty in achieving myocardial nulling was 100%, with a sensitivity of 68%.”

Table 3

Frequencies of positive diagnostic criteria in the HCM and CA patient groups compared with the chi-squared test

We have updated the manuscript’s style according to PLOS ONE’s style requirements.

We have uploaded our study’s minimal underlying data set as Supporting Information file.

“Project no. NVKP_16-1–2016-0017 (’National Heart Program’) has been implemented with the support provided from the National Research, Development and Innovation Fund of Hungary, financed under the NVKP_16 funding scheme. The research was financed by the Thematic Excellence Programme (2020-4.1.1.-TKP2020) of the Ministry for Innovation and Technology in Hungary, within the framework of the Therapeutic Development and Bioimaging thematic programmes of the Semmelweis University; and by the Ministry of Innovation and Technology NRDI Office within the framework of the Artificial Intelligence National Laboratory Program. LS was supported by the ÚNKP-20-3-II-SE-61 New National Excellence Program of the Ministry for Innovation and Technology from the source of the National Research, Development and Innovation Fund. ZD and LS were supported by the „Development of scientific workshops of medical, health sciences and pharmaceutical educations” project. Project identification number: EFOP-3.6.3-VEKOP-16-2017-00009.

We note that you have provided additional information within the Funding Section. Please note that funding information should not appear in other areas of your manuscript. We will only publish funding information present in the Funding Statement section of the online submission form.

“Project no. NVKP_16-1–2016-0017 (’National Heart Program’) has been implemented with the support provided from the National Research, Development and Innovation Fund of Hungary, financed under the NVKP_16 funding scheme. The research was financed by the Thematic Excellence Programme (2020-4.1.1.-TKP2020) of the Ministry for Innovation and Technology in Hungary, within the framework of the Therapeutic Development and Bioimaging thematic programmes of the Semmelweis University; and by the Ministry of Innovation and Technology NRDI Office within the framework of the Artificial Intelligence National Laboratory Program. LS was supported by the ÚNKP-20-3-II-SE-61 New National Excellence Program of the Ministry for Innovation and Technology from the source of the National Research, Development and Innovation Fund. ZD and LS were supported by the „Development of scientific workshops of medical, health sciences and pharmaceutical educations” project. Project identification number: EFOP-3.6.3-VEKOP-16-2017-00009. The funders had no role in study design, data collection and analysis, decision to publish, or preparation of the manuscript.”

We have removed funding-related text from the manuscript. We have not changed our Funding Statement.

We have included captions for the Supporting Information file at the end of the manuscript.

5) In order to refine and focus the findings, please exclude patients with Fabry disease and endomyocardial fibrosis.

Thank you for your advice. We excluded patients with Fabry disease and endomyocardial fibrosis from the study.

Reviewers' comments:

Author comment:

We would like to thank the reviewer for the thorough reading of our manuscript and the constructive suggestions. We believe that the manuscript has been improved by incorporating these comments and recommendations.

Reviewer #1: 

In the present paper Dohy and Merkely et al investigate the ability of myocardial strain by feature tracking analysis for the diagnostic classification and risk stratification of patients with myocardial hypertrophy due to HCM, cardiac amyloidosis (AL and ATTR), Fabry disease and endomyocardial fibrosis. Myocardial strain variables were able to differentiate between HCM and cardiac amyloidosis, whereas risk stratification could also be provided. The paper is surely of interest since strain is an important issue with cardiac imaging and the differentiation between diseases causing hypertrophy has nowadays important therapeutic implications for the specific management of such patients more than ever. However, some specific points definitely need to be clarified.

1. The percentage of patients with HCM is strikingly high compared to those with cardiac amyloidosis. Especially ATTR patients are expected to be found much more frequently in a tertiary setting and AL patients in conjunction with hematology / oncology units. Please explain and consider reporting / acknowledging referral biases.

2. In the same direction, the number of ATTR patients is strikingly low with 3 patients in total! This is certainly not in agreement with current trends of diagnosis this entity in 30-fold more patients than in the last decade. The prevalence especially in elderly patients with hypertrophy of unknown origin is very high, if correctly diagnosed based on current algorithms.

Thank you for your questions.

Answer for question 1 and 2:

There was certainly some referral bias. Myeloma and MGUS pts are monitored for CA in hematology centers, from where a great number of AL patients are referred. In other words, hematologists have a high awareness of AL. For them, CMR is ideal for the diagnosis of CA.

On the other hand, ATTR patients are usually referred from community cardiac services or from other cardiology centers where diagnostic awareness is lower (see below). In recent years, these centers have diagnosed TTR CA in a "non-biopsy" manner in many cases with the help of PYP scans. Many of these TTR CA pts did not have a CMR for the diagnosis. Another possible explanation for the relatively low number of ATTRwt patients is that the expected life expectancy in Hungary is significantly lower than that in Western Europe and the US, and therefore fewer patients "live long enough" to obtain ATTRwt. Additionally, the severe comorbidity of the elderly population in Hungary is quite high; therefore, we speculate that CMR is not performed in many cases, as it would not change the disease course.

In a very large series of systemic amyloid (not only CA) pts, Wechalekar et al published in 2016 in the Lancet that the ratio of AL was stable at approximately 67% over the past decades. The ratio of ATTR was growing simultaneously with the lowering number of AAs. In our retrospective study, the cause of CA remained unknown in 7 pts, but probably some of them also had ATTR. All together our data, i.e., the ratio of AL and ATTR, are not far from the data of Wechalekar et al.

CA is considered to be a rare disease, and diagnostic awareness is still low in Hungary. It is nicely shown by a recent paper which found that the prevalence of ATTRv is just half of the prevalence observed in non-endemic regions of Western Europe, although the number of recognized cases is increasing. (Pozsonyi et al, 2021, Genes). This is also one cause, which explains the higher ratio of HCM and lower CA patients among LVH morphology pts in Hungary.

We have completed the limitations sections with the followings: “There might be some referral bias that can explain the relatively low prevalence of CA patients, especially ATTR patients: ATTR patients are usually referred from community cardiac services or from other cardiology centers where diagnostic awareness is lower compared to hematology centers where myeloma and MGUS patients are monitored for CA and from where a great number of AL patients are referred. In recent years, cardiology centers have diagnosed TTR CA in a "non-biopsy" manner in many cases with the help of PYP scans. Many of these TTR CA pts did not have a CMR for the diagnosis.”

3. In the same line the authors did not include patients with hypertensive heart disease, where in many cases differential diagnosis regarding HCM and cardiac amyloidosis may be very challenging, especially in those with progressed hypertensive heart disease, where focal LGE may be present, mimicking HCM. This needs to be explained by the authors, since inclusion of such patients would have been clinically meaningful and since inclusion of patients has performed retrospectively, which means that such patients could have been easily included and considered for feature tracking analysis too.

Thank you for your comment. Differentiating hypertensive heart disease from HCM is highly clinically relevant. Patients with left ventricular hypertrophy obviously due to untreated hypertension are usually not referred to CMR in our center. Patients with hypertensive heart disease are usually referred for CMR if there is a suspicion of other underlying conditions behind LV hypertrophy. In these cases, differentiating hypertensive heart disease from mild phenotypic HCM is challenging. To avoid diagnostic uncertainty, we decided to exclude these patients from the study. As we involved in the study referred patients with clinical questions, we had only a few cases of LV hypertrophy certainly due to hypertensive heart disease.

4. The same restrictions also apply for individuals with athletes’ heart, while in this case referral may be an issue.

Thank you for your comment. We agree with the reviewer that differentiating athletes’ heart and pathological hypertrophy is crucial. As our clinic is a sports cardiology center, we perform CMR examinations in athletes in a large number. Nine athletes who were diagnosed with HCM were involved in the study. Athletes with physiological sport adaptation were not investigated in the current study.

We completed the methods section with the followings: “Patients with untreated hypertension, significant aortic stenosis, or athletes with left ventricular hypertrophy due to physiological sport adaptation were not involved in the study.”

5. Figure 1. It is striking that patients with hypertensive heart disease are not included. Please see also my comment #3.

Thank you for your comment. According to our inclusion criteria, patients with hypertensive heart disease were not included in the study.

6. Figure 2. It would be meaningful to parallel show the images of a normal volunteer for comparison to this cardiac amyloidosis patient with very progressed myocardial disease.

Thank you for your advice. We added to Figure 2 the LGE images and the bull’s eye with segmental strain values of a patient without structural heart disease.

7. Diagnosis of HCM. What about patients with septal wall thickness >13mm and family history of HCM?

Thank you for your question. The diagnosis of HCM in first-degree relatives of patients with HCM was based on the presence of otherwise unexplained increased wall thickness ≥13 mm. We have completed the methods section with this information. HCM in the family history was known in 24 cases, and only 4 of them had a wall thickness of 13-14 mm. 

8. I am not sure if inclusion of patients with endomyocardial fibrosis is meaningful since hypertrophy is not necessarily associated with this entity. Possibly some overlap with hypertensive heart disease needs to be considered with these patients more than with other categories.

Thank you for your comment. On the advice of the Editor, patients with EMF or Fabry disease have been excluded from the study.

9. The differential diagnosis of HCM versus cardiac amyloidosis is clinical meaningful and a strength of the article. It is correct to highlight this in the results, as shown in Table 2. Hereby, LGE seems to be the stronger predictor, which can be expected. However, the value of strain is also relevant since basal LS und CS also provide relatively high AUC values.

Thank you for your comment. We have completed this section with the followings:

“The frequencies of positive results of different diagnostic criteria in the patient groups are shown in Table 3. The presence of septal or septal and posterior diffuse subendocardial LGE had high specificity (99% and 99.4%, respectively) and relatively high sensitivity (89% for both). The specificity of myocardial nulling prior to blood pool nulling or difficulty in achieving myocardial nulling was 100%, with a sensitivity of 68%.”

Table 3 Frequencies of positive diagnostic criteria in the HCM and CA patient groups compared with the chi-squared test 

We have highlighted these results in the discussion section:

“In the differentiation of CA from HCM, the amount and pattern of LGE have the highest diagnostic accuracy, followed by basal LS, basal CS and GRS. (…) In our study population, it was found that the amount and pattern of LGE had the highest diagnostic accuracy in the differentiation of CA and HCM. However, in case of a contraindication for contrast agent administration, further diagnostic methods are needed. We found that strain parameters have relatively high diagnostic accuracy. The sensitivity of GLS with a cut-off of -23% was 89%, while the specificities of basal LS (cut-off of -16%), basal CS (cut-off of -31%) and GRS (cut-off of 63%) were 85%, 83% and 83%, respectively.”

10. Figure 4 demonstrates that patients with amyloidosis have poorer prognosis, which is expected due to the entity of the disease. However, much more meaningful would be a Kaplan-Maier analysis based on LGE and on strain values. Would one of the myocardial strain values be able to show prognostic relevance. If yes, this would surely contribute to the current literature.

Thank you for your advice. Using ROC curve analysis, we calculated cut-off values of LGE, LVSVi, GLS, GCS and GRS regarding mortality. The survival probability of patient groups defined by the calculated cut-off parameters was analyzed and illustrated with Kaplan-Meier curves, which can be found in the revised Figure 4.

11. In the same direction, basal LS seems to bear independent prognostic implications for the estimation of mortality. This needs to be tested and adjusted for quantitative LGE and main diagnosis. The corresponding Kaplan-Maier analysis needs to be demonstrated.

Thank you for your comment. Our study population changed, as we excluded patients with EMF or Fabry disease from the analysis on the advice of the Editor. Furthermore, we completed the regional strain analysis of all HCM patients (in the original manuscript, regional strain analysis was performed on 89 randomly selected HCM patients). These factors led to a changed result of the survival analysis: independent predictors of mortality with multivariable Cox proportional hazard regression are a diagnosis of CA, age and GLS. Variables with p<0.05 in the univariable analysis were analyzed in one multivariable model (after excluding highly correlated predictors). As quantitative LGE and the main diagnosis were included in this model, the prognostic significance of GLS is independent of these factors. We believe this result is more reliable than the previous one was with a smaller patient group.

Kaplan-Meier curves based on LGE, LVSVi, GLS, GCS and GRS have been added to Figure 4.

12. A recent study investigating the ability of strain based CMR (with fast-SENC) to differentiate between patients with HCM, amyloidosis and hypertensive heart disease needs to be reported and discussed by the authors (Giusca et al, JCMR 2021).

Thank you for your comment. We have added the required article to the references and completed the discussion section with the followings: “A recently published study investigated the ability of a single heartbeat fast-strain encoded (SENC) CMR-derived myocardial strain to discriminate between CA, HCM, hypertensive heart disease, athletes’ heart and healthy controls. CA patients had the most impaired GLS and GCS values, and the percentage of LV segments with a strain value < -17% was the lowest in this patient group; apical sparing was not investigated.”

13. In addition, some limitations need to be reported with feature tracking acquisitions. Although, a big advantage of feature tracking is no need for additional dedicated CMR sequences, some disadvantages have been extensively mentioned in the recent literature, especially regarding the segmental assessment of strain (Feisst A et al, IJC Heart Vasc. 2018; Mangion K et al, Sci Rep. 2019 and Almitairi HM et al, Br. J. Radiol. 2017).

Thank you for your comment. We have completed the limitations section as follows: “The main advantage of feature-tracking strain analysis is that it needs no additional dedicated CMR sequences, and the evaluation is performed using the standard cine images. However, this method has some limitations: previously published data showed that reliability and accuracy of feature-tracking analysis is dependent on reader experience more than tagging-based strain analysis, and the reproducibility of segmental assessment of strain is lower.”

---

## [Editor Report · Decision Letter 1]

25 Nov 2021

PONE-D-21-24457R1The role of cardiac magnetic resonance-based feature-tracking strain analysis in the differential diagnosis and prognostic assessment of patients with left ventricular hypertrophy.PLOS ONE

Dear Dr. Vago,

Thank you for submitting your manuscript to PLOS ONE. After careful consideration, we feel that it has merit but does not fully meet PLOS ONE’s publication criteria as it currently stands. Therefore, we invite you to submit a revised version of the manuscript that addresses the points raised during the review process.

Please include the following items when submitting your revised manuscript:A rebuttal letter that responds to each point raised by the academic editor and reviewer(s). You should upload this letter as a separate file labeled 'Response to Reviewers'.A marked-up copy of your manuscript that highlights changes made to the original version. You should upload this as a separate file labeled 'Revised Manuscript with Track Changes'.An unmarked version of your revised paper without tracked changes. You should upload this as a separate file labeled 'Manuscript'.

We look forward to receiving your revised manuscript.

Kind regards,

Daniel A. Morris, M.D

Academic Editor

PLOS ONE

 Editor Comments :

I have read and revised again this interesting large study and appreciated the effort of the authors to address the suggestions of the editors and reviewers. While the data is interesting, there are pending major and serious limitations in this study, mainly regarding the clinical relevance, applicability, and presentation of this study.

Hence, without addressing the below detailed major limitations, this study will have a low priority for publication in PlosOne.

Major and Serious Pending Limitations:

1) Presentation of the Manuscript and Results:

- The most important and clinically relevant findings from this study are those linked to AL-Amyloidosis (i.e., the sensitivity and specificity of some parameters to detect a cardiac involvement in this systematic hematological disease). In fact, while patients with cardiac ATTR-Amyloidosis are easily diagnosed with bone scintigraphy (planar and/or SPECT), the cardiac involvement in AL-Amyloidosis is challenging (i.e., SPECT does not provide any diagnostic help). In addition, the role of CMR in AL-Amyloidosis remains uncertain. Hence, the authors should make focus in patients with AL-Amyloidosis.

- In line with the above-mentioned comments, it will necessary to include a control group of at least 70 patients (i.e., at least match 1:2) with arterial hypertension without history of cardiomyopathy and with similar LVEF than the group with AL-Amyloidosis. Including merely at least 70 control patients it would not mean a lot of effort in a normal CMR department.

- Patients with other type of Amyloidosis other than AL should be excluded in order to get a homogenous and clinically relevant population.

- The most important analyses and parameters to analyze are the sensibility and specificity of CMR parameters to determine cardiac involvement as compared to controls patients and also to differentiate from CMH. Hence, the authors should mandatorily analyze and show the sensibility and specificity of the following parameters to determine cardiac Al-Amyloidosis (namely, in 2 separated tables, one AL-Amyloidosis vs. arterial hypertension; and another Al-Amyloidosis vs. CMH):

Septal and posterior wall thickness ≥ 15mm

Septal and posterior wall thickness ≥ 14mm

Septal and posterior wall thickness ≥ 13mm

Septal and posterior wall thickness ≥ 12mm

Septal wall thickness ≥ 20mm

Septal wall thickness ≥ 18mm

Septal wall thickness ≥ 15mm

Septal wall thickness ≥ 14mm

Septal wall thickness ≥ 13mm

Septal wall thickness ≥ 12mm

Septal and posterior diffuse subendocardial LGE

Septal diffuse subendocardial LGE

Septal and posterior myocardial nulling prior to blood pool nulling or difficulty in achieving myocardial nulling

Septal myocardial nulling prior to blood pool nulling or difficulty in achieving myocardial nulling

Apical sparing using LV global longitudinal strain (GLS) (i.e., ratio average apical segments to average basal segments > 2).

GLS < 15% and apical sparing

GLS < 15%

GLS < 13%

GLS < 12%

- Please provide the following cases examples:

1- AL-Amyloidosis with septal and posterior myocardial nulling prior to blood pool nulling or difficulty in achieving myocardial nulling

2- Arterial hypertension without septal and posterior myocardial nulling prior to blood pool nulling or difficulty in achieving myocardial nulling

3- CMH without septal and posterior myocardial nulling prior to blood pool nulling or difficulty in achieving myocardial nulling

4- AL-Amyloidosis with septal and posterior diffuse subendocardial LGE

5- Arterial hypertension without septal and posterior diffuse subendocardial LGE

6- CMH without septal and posterior diffuse subendocardial LGE

- The outcome data is not of high relevance in this study, but if you consider important this data, please analyze only the mortality for HF or hospitalization for HF and in separated groups (i.e., those with CMH, AL-Amyloidosis, and arterial hypertension).

---

## [Author Response · Author response to Decision Letter 1]

13 Dec 2021

Editor Comments :

I have read and revised again this interesting large study and appreciated the effort of the authors to address the suggestions of the editors and reviewers. While the data is interesting, there are pending major and serious limitations in this study, mainly regarding the clinical relevance, applicability, and presentation of this study.

Hence, without addressing the below detailed major limitations, this study will have a low priority for publication in PlosOne.

Major and Serious Pending Limitations:

1) Presentation of the Manuscript and Results:

- The most important and clinically relevant findings from this study are those linked to AL-Amyloidosis (i.e., the sensitivity and specificity of some parameters to detect a cardiac involvement in this systematic hematological disease). In fact, while patients with cardiac ATTR-Amyloidosis are easily diagnosed with bone scintigraphy (planar and/or SPECT), the cardiac involvement in AL-Amyloidosis is challenging (i.e., SPECT does not provide any diagnostic help). In addition, the role of CMR in AL-Amyloidosis remains uncertain. Hence, the authors should make focus in patients with AL-Amyloidosis.

Thank you for your comment. We have focused on patients with AL-Amyloidosis and have excluded patients with TTR or AA amyloidosis or if the exact type of amyloidosis was unknown.

- In line with the above-mentioned comments, it will necessary to include a control group of at least 70 patients (i.e., at least match 1:2) with arterial hypertension without history of cardiomyopathy and with similar LVEF than the group with AL-Amyloidosis. Including merely at least 70 control patients it would not mean a lot of effort in a normal CMR department.

Thank you for your advice. Hypertensive heart disease patients are usually not referred for CMR in our Center if there is no suspicion of cardiomyopathy. We can retrospectively collect patients, who have in their patient’s history arterial hypertension (mostly treated hypertension) and were referred for CMR with an other indication but CMR found no structural heart disease. However, these patients usually have no expressed hypertrophy as their hypertension is treated. If the Editor suggest that we could increase the scientific value of our study with analyzing this patient population, we will complete our study with pleasure.

- Patients with other type of Amyloidosis other than AL should be excluded in order to get a homogenous and clinically relevant population.

Thank you for your advice. We have excluded patients with TTR or AA amyloidosis and if the exact type of amyloidosis was unknown.

- The most important analyses and parameters to analyze are the sensibility and specificity of CMR parameters to determine cardiac involvement as compared to controls patients and also to differentiate from CMH. Hence, the authors should mandatorily analyze and show the sensibility and specificity of the following parameters to determine cardiac Al-Amyloidosis (namely, in 2 separated tables, one AL-Amyloidosis vs. arterial hypertension; and another Al-Amyloidosis vs. CMH):

Septal and posterior wall thickness ≥ 15mm

Septal and posterior wall thickness ≥ 14mm

Septal and posterior wall thickness ≥ 13mm

Septal and posterior wall thickness ≥ 12mm

Septal wall thickness ≥ 20mm

Septal wall thickness ≥ 18mm

Septal wall thickness ≥ 15mm

Septal wall thickness ≥ 14mm

Septal wall thickness ≥ 13mm

Septal wall thickness ≥ 12mm

Septal and posterior diffuse subendocardial LGE

Septal diffuse subendocardial LGE

Septal and posterior myocardial nulling prior to blood pool nulling or difficulty in achieving myocardial nulling

Septal myocardial nulling prior to blood pool nulling or difficulty in achieving myocardial nulling

Apical sparing using LV global longitudinal strain (GLS) (i.e., ratio average apical segments to average basal segments > 2).

GLS < 15% and apical sparing

GLS < 15%

GLS < 13%

GLS < 12%

Thank you for your advice. We have added a table with the sensitivity and specificity of the required parameters to differentiate CA from HCM.

- Please provide the following cases examples:

1- AL-Amyloidosis with septal and posterior myocardial nulling prior to blood pool nulling or difficulty in achieving myocardial nulling

2- Arterial hypertension without septal and posterior myocardial nulling prior to blood pool nulling or difficulty in achieving myocardial nulling

3- CMH without septal and posterior myocardial nulling prior to blood pool nulling or difficulty in achieving myocardial nulling

4- AL-Amyloidosis with septal and posterior diffuse subendocardial LGE

5- Arterial hypertension without septal and posterior diffuse subendocardial LGE

6- CMH without septal and posterior diffuse subendocardial LGE

Thank you for your comment. We provided examples of CA with myocardial nulling prior to blood pool nulling and diffuse subendocardial LGE, and a case of HCM with patchy mid-myocardial LGE in Figure 3.

- The outcome data is not of high relevance in this study, but if you consider important this data, please analyze only the mortality for HF or hospitalization for HF and in separated groups (i.e., those with CMH, AL-Amyloidosis, and arterial hypertension).

Thank you for your advice. We have no information about the hospitalization in most of the cases, and also the cause of death is unknown in a part of the cases. Therefore, we decided to leave out the Cox regression analysis for the assessment the prognostic value of CMR parameters.

The Kaplan-Meier analyses are included the study later on, as it was the request of the Reviewer.

---

## [Editor Report · Decision Letter 2]

15 Dec 2021

PONE-D-21-24457R2The role of cardiac magnetic resonance-based feature-tracking strain analysis in the differential diagnosis and prognostic assessment of patients with left ventricular hypertrophy.PLOS ONE

Dear Dr. Vago,

Thank you for submitting your manuscript to PLOS ONE. After careful consideration, we feel that it has merit but does not fully meet PLOS ONE’s publication criteria as it currently stands. Therefore, we invite you to submit a revised version of the manuscript that addresses the points raised during the review process.

Please include the following items when submitting your revised manuscript:A rebuttal letter that responds to each point raised by the academic editor and reviewer(s). You should upload this letter as a separate file labeled 'Response to Reviewers'.A marked-up copy of your manuscript that highlights changes made to the original version. You should upload this as a separate file labeled 'Revised Manuscript with Track Changes'.An unmarked version of your revised paper without tracked changes. You should upload this as a separate file labeled 'Manuscript'.

We look forward to receiving your revised manuscript.

Kind regards,

Daniel A. Morris, M.D

Academic Editor

PLOS ONE

Additional Editor Comments:

I would like to congratulate to the authors for the effort made to improve the manuscript following the suggestions of the Editors and Reviewers. In fact, the presentation of the study has significantly improved and it remains just some limitations to be addressed to get the final version of this interesting study.

Minor Pending Limitations:

1) It will absolutely necessary to include a control group of at least 70 patients (i.e., at least match 1:2) with arterial hypertension without history of cardiomyopathy and with similar LVEF than the group with AL-Amyloidosis.

2) Please a table as table 3, comparing the sensibility and specificity of the parameters of table but comparing AL-Amyloidosis vs. control patients with arterial hypertension.

3) Table 2 is not interesting and thus, it should be moved to data supplement.

4) The section on prognosis is interesting, but out of the scope of the present study. Hence, it should be removed, but it would be very interesting to make a new study and analysis examining the main parameters linked to prognosis in patients with CMH in this large cohort of 330 patients. PlosOne Editors will evaluate with interest this potential further study.

5) The most important and clinically relevant findings from this study are those linked to AL-Amyloidosis (i.e., the sensitivity and specificity of some parameters to detect a cardiac involvement in this systematic hematological disease). In fact, while patients with cardiac ATTR-Amyloidosis are easily diagnosed with bone scintigraphy (planar and/or SPECT), the cardiac involvement in AL-Amyloidosis is challenging (i.e., SPECT does not provide any diagnostic help). In addition, the role of CMR in AL-Amyloidosis remains uncertain. Hence, the authors should make focus in patients with AL-Amyloidosis.

6) In the line with above-mentioned suggestion, the authors should significantly change the title, abstract, introduction, and discussion of the study, focusing only in the findings on the detection and diagnosis of cardiac AL-Amyloidosis. In this respect, a potential title would be “Potential Clinical Relevance of CMR to Diagnose Cardiac AL-Amyloidosis”. By the way, a potential abstract would be:

- Background: While patients with cardiac ATTR-Amyloidosis are easily diagnosed with bone scintigraphy (planar and/or SPECT), the detection of cardiac involvement in AL-Amyloidosis is challenging. In addition, the role of CMR in AL-Amyloidosis remains uncertain. Hence, the purpose of the present study was to analyze the potential role of CMR in the detection of cardiac involvement in patients with AL-Amyloidosis.

Methods: We included 35 patients with proved cardiac AL-Amyloidosis and two control groups constituted by 330 patients with HCM and 70 patients with arterial hypertension, who underwent CMR examination. Phenotype and amount of LV wall thickness, strain, and late gadolinium enhancement (LGE) were evaluated. Sensibility and specificity of several CMR parameters were analyzed comparing patients with cardiac AL-Amyloidosis vs those with CMH and vs. those with arterial hypertension.

Results: please describe the results of table 3 regarding AL-Amyloidosis vs CMH and vs. arterial hypertension.

Conclusions: The findings from this study suggest that septal myocardial nulling prior to blood pool nulling or difficulty in achieving myocardial nulling or septal diffuse subendocardial LGE proving excellent sensibility and specificity to determine cardiac involvement in patients with AL-Amyloidosis. Further larger studies are warranted to validate the findings from this study.

7) Please do not add the symbol % next to the parameters’ description, but next to the values of sensibility and specificity.

8) Please add to the table 3 a file with the accuracy value.

9) Please provide mandatorily the following cases/figures examples:

1- AL-Amyloidosis with septal myocardial nulling prior to blood pool nulling or difficulty in achieving myocardial nulling

2- Arterial hypertension without septal myocardial nulling prior to blood pool nulling or difficulty in achieving myocardial nulling

3- CMH without septal myocardial nulling prior to blood pool nulling or difficulty in achieving myocardial nulling

4- AL-Amyloidosis with septal diffuse subendocardial LGE

5- Arterial hypertension without septal diffuse subendocardial LGE

6- CMH without septal diffuse subendocardial LG

10) Please provide a table comparing and describing the findings of the present study vs previous similar studies (i.e., role of CMR in AL-Amyloidosis; please see https://pubmed.ncbi.nlm.nih.gov/?term=light%20chain%20(AL)%20amyloidosis%20AND%20diagnosis%20AND%20CMR&sort=date&page=4

11) Please discuss previous similar studies in the discussion section.

---

## [Author Response · Author response to Decision Letter 2]

28 Jan 2022

1) It will absolutely necessary to include a control group of at least 70 patients (i.e., at least match 1:2) with arterial hypertension without history of cardiomyopathy and with similar LVEF than the group with AL-Amyloidosis.

Thank you for your suggestion. We have added 70 patients with arterial hypertension to the study and analyzed the differential diagnostic significance of CMR parameters in CA patients vs. those with hypertension. We believe that the clinical relevance of our study increased with these findings.

2) Please a table as table 3, comparing the sensibility and specificity of the parameters of table but comparing AL-Amyloidosis vs. control patients with arterial hypertension.

Thank you for your suggestion. We have added a table with the sensitivity and specificity of CMR parameters to differentiate CA from controls with HT

3) Table 2 is not interesting and thus, it should be moved to data supplement.

Thank you for your advice. We have moved the results of ROC analysis to data supplement.

4) The section on prognosis is interesting, but out of the scope of the present study. Hence, it should be removed, but it would be very interesting to make a new study and analysis examining the main parameters linked to prognosis in patients with CMH in this large cohort of 330 patients. PlosOne Editors will evaluate with interest this potential further study.

Thank you for your suggestion. We have removed prognosis from the current study, and we will consider to publish it in a further study.

5) The most important and clinically relevant findings from this study are those linked to AL-Amyloidosis (i.e., the sensitivity and specificity of some parameters to detect a cardiac involvement in this systematic hematological disease). In fact, while patients with cardiac ATTR-Amyloidosis are easily diagnosed with bone scintigraphy (planar and/or SPECT), the cardiac involvement in AL-Amyloidosis is challenging (i.e., SPECT does not provide any diagnostic help). In addition, the role of CMR in AL-Amyloidosis remains uncertain. Hence, the authors should make focus in patients with AL-Amyloidosis.

Thank you for your comment. We have made changes in the introduction and discussion section in order to direct the focus on the diagnosis of AL-amyloidosis. 

6) In the line with above-mentioned suggestion, the authors should significantly change the title, abstract, introduction, and discussion of the study, focusing only in the findings on the detection and diagnosis of cardiac AL-Amyloidosis. In this respect, a potential title would be “Potential Clinical Relevance of CMR to Diagnose Cardiac AL-Amyloidosis”. By the way, a potential abstract would be:

- Background: While patients with cardiac ATTR-Amyloidosis are easily diagnosed with bone scintigraphy (planar and/or SPECT), the detection of cardiac involvement in AL-Amyloidosis is challenging. In addition, the role of CMR in AL-Amyloidosis remains uncertain. Hence, the purpose of the present study was to analyze the potential role of CMR in the detection of cardiac involvement in patients with AL-Amyloidosis.

Methods: We included 35 patients with proved cardiac AL-Amyloidosis and two control groups constituted by 330 patients with HCM and 70 patients with arterial hypertension, who underwent CMR examination. Phenotype and amount of LV wall thickness, strain, and late gadolinium enhancement (LGE) were evaluated. Sensibility and specificity of several CMR parameters were analyzed comparing patients with cardiac AL-Amyloidosis vs those with CMH and vs. those with arterial hypertension.

Results: please describe the results of table 3 regarding AL-Amyloidosis vs CMH and vs. arterial hypertension.

Conclusions: The findings from this study suggest that septal myocardial nulling prior to blood pool nulling or difficulty in achieving myocardial nulling or septal diffuse subendocardial LGE proving excellent sensibility and specificity to determine cardiac involvement in patients with AL-Amyloidosis. Further larger studies are warranted to validate the findings from this study.

Thank you for your suggestion. We have changed the title and the abstract of the article. We agree with the Editor that LGE CMR has an important role in the diagnosis of cardiac AL-amyloidosis. Moreover, we believe that our findings regarding the CMR-based strain analysis have also clinical significance and novelty. Therefore, besides LGE results, we would like to focus on strain, as well.

Full title: Potential clinical relevance of cardiac magnetic resonance including strain analysis to diagnose cardiac light chain amyloidosis.

Short title: CMR-based diagnosis of patients with cardiac amyloidosis.

Abstract:

Background: While patients with cardiac transthyretin amyloidosis are easily diagnosed with bone scintigraphy (planar and/or SPECT), the detection of cardiac involvement in light chain amyloidosis (CA) is challenging. Cardiac magnetic resonance (CMR) examinations have an essential role in the diagnosis of myocardial diseases; however, limited data are available from CMR-based feature-tracking strain analysis in this patient population. Hence, the purpose of the present study was to analyze the potential role of CMR in the detection of cardiac involvement in patients with light chain amyloidosis (CA).

Methods: We included 35 patients with proved CA and two control groups constituted by 330 patients with hypertrophic cardiomyopathy (HCM) and 70 patients with arterial hypertension (HT), who underwent CMR examination. The phenotype and degree of left ventricular (LV) hypertrophy, and the amount and pattern of late gadolinium enhancement (LGE) were evaluated. Global and regional LV strain parameters were calculated with feature-tracking strain analysis. Sensitivity and specificity of several CMR parameters were analyzed in diagnosing CA.

Results: The sensitivity of diffuse septal subendocardial LGE in diagnosing CA was 88% with a specificity of 99% when differentiating it from HCM and 100% when differentiating it from HT. The sensitivity and specificity of myocardial nulling prior to blood pool was 77% and 100%, respectively, both vs. HCM and vs. HT. Basal longitudinal strain had also high diagnostic accuracy (CA vs. HCM: sensitivity 69%, specificity 85%; CA vs. HT: sensitivity 94%, specificity 79%). CMR-based strain analysis was applicable for detecting apical sparing in CA patients.

Conclusions: The findings from this study suggest that CMR has high diagnostic relevance in the diagnosis of CA. Besides the excellent sensitivity and specificity of diffuse subendocardial LGE pattern and abnormal contrast kinetics, CMR-based strain analysis has also an important role in differentiating CA from HCM and from controls with HT.

7) Please do not add the symbol % next to the parameters’ description, but next to the values of sensibility and specificity.

Thank you for your advice. We have added the symbol % next to the values instead of the parameters’ description.

8) Please add to the table 3 a file with the accuracy value.

Thank you for your advice. We added the AUC values to the table 3.

9) Please provide mandatorily the following cases/figures examples:

1- AL-Amyloidosis with septal myocardial nulling prior to blood pool nulling or difficulty in achieving myocardial nulling

2- Arterial hypertension without septal myocardial nulling prior to blood pool nulling or difficulty in achieving myocardial nulling

3- CMH without septal myocardial nulling prior to blood pool nulling or difficulty in achieving myocardial nulling

4- AL-Amyloidosis with septal diffuse subendocardial LGE

5- Arterial hypertension without septal diffuse subendocardial LGE

6- CMH without septal diffuse subendocardial LG

Thank you for your suggestion. On Figure 3, we show a representative case of HCM without septal myocardial nulling prior to blood pool nulling or difficulty in achieving myocardial nulling and without septal diffuse subendocardial LGE (A), cases of CA with myocardial nulling prior to blood pool nulling (B) and diffuse subendocardial LGE (C), and a case of arterial hypertension without septal myocardial nulling prior to blood pool nulling or difficulty in achieving myocardial nulling and without septal diffuse subendocardial LGE (D).

10) Please provide a table comparing and describing the findings of the present study vs previous similar studies (i.e., role of CMR in AL-Amyloidosis; please see https://pubmed.ncbi.nlm.nih.gov/?term=light%20chain%20(AL)%20amyloidosis%20AND%20diagnosis%20AND%20CMR&sort=date&page=4

Thank you for your advice. Recently a meta-analysis was published investigating the diagnostic performance of CMR in cardiac amyloidosis based on all relevant studies, thus we decided to refer to this article, as we cannot perform a more detailed analysis regarding this topic.

11) Please discuss previous similar studies in the discussion section.

Thank you for your suggestion. We have completed the discussion section with the followings:

“A recent meta-analysis based on 18 published studies included 1,108 CA patients (69% were AL) and 907 control subjects, estimated a sensitivity and specificity of 84% and 80%, respectively, for LGE in diagnosing CA (21). According to another meta-analysis of 7 studies, the sensitivity and specificity of LGE CMR in diagnosing CA were 85% and 92%, respectively (22). Expert consensus recommendations (17,18) state that CMR has a central role in the non-invasive diagnosis of CA referring to several studies in which typical LGE pattern has been shown to have a diagnostic sensitivity of 85% to 90% (23–27).”

---

## [Editor Report · Decision Letter 3]

1 Mar 2022

PONE-D-21-24457R3Potential clinical relevance of cardiac magnetic resonance including strain analysis to diagnose cardiac light chain amyloidosis.PLOS ONE

Dear Dr. Vago,

Thank you for submitting your manuscript to PLOS ONE. After careful consideration, we feel that it has merit but does not fully meet PLOS ONE’s publication criteria as it currently stands. Therefore, we invite you to submit a revised version of the manuscript that addresses the points raised during the review process.

Please include the following items when submitting your revised manuscript:A rebuttal letter that responds to each point raised by the academic editor and reviewer(s). You should upload this letter as a separate file labeled 'Response to Reviewers'.A marked-up copy of your manuscript that highlights changes made to the original version. You should upload this as a separate file labeled 'Revised Manuscript with Track Changes'.An unmarked version of your revised paper without tracked changes. You should upload this as a separate file labeled 'Manuscript'.

We look forward to receiving your revised manuscript.

Kind regards,

Daniel A. Morris, M.D

Academic Editor

PLOS ONE

Additional Editor Comments:

Thank you very much for your time in addressing all suggested revisions. In effect, the paper has significantly improved and it is still only some minor revisions to get the final version of this interesting study.

Pending Minor Revisions:

1) Please use the term Al-Amyloidosis to refer to light chain amyloidosis. Hence, please use in the whole manuscript as well in the abstract and conclusion the term “cardiac AL-Amyloidosis”.

2) Please make focus in the results section of the abstract and in the conclusion of the manuscript on the main findings of the study such as the high sensitivity and specificity of diffuse septal subendocardial LGE and myocardial nulling to differentiate cardiac AL-Amyloidosis from HCM and from HT as well as the low sensibility of a reduced GLS and of apical sparing to differentiate cardiac AL-Amyloidosis from HCM and from HT. Moreover, please highlight that a septum ≥ 14 mm has high sensitivity and specificity to differentiate cardiac AL-Amyloidosis from HT.

3) The results on the diagnostic performance of LV basal segmental strain are additional or secondary findings given the high variability and low reproducibility of basal strain values in the LV. Hence, the manuscript should not be focused or centered on these findings and these findings should not be shown in the abstract or conclusion section.

4) In line with the above-mentioned comments, please re-edit the title of the study as “Potential clinical relevance of cardiac magnetic resonance to diagnose cardiac light chain amyloidosis”.

---

## [Author Response · Author response to Decision Letter 3]

4 Apr 2022

1) Please use the term Al-Amyloidosis to refer to light chain amyloidosis. Hence, please use in the whole manuscript as well in the abstract and conclusion the term “cardiac AL-Amyloidosis”.

Thank you for your suggestion. We have changed the term CA to cardiac AL-amyloidosis.

2) Please make focus in the results section of the abstract and in the conclusion of the manuscript on the main findings of the study such as the high sensitivity and specificity of diffuse septal subendocardial LGE and myocardial nulling to differentiate cardiac AL-Amyloidosis from HCM and from HT as well as the low sensibility of a reduced GLS and of apical sparing to differentiate cardiac AL-Amyloidosis from HCM and from HT. Moreover, please highlight that a septum ≥ 14 mm has high sensitivity and specificity to differentiate cardiac AL-Amyloidosis from HT.

Thank you for your advice. We have changed the results section of the abstract as follows:

“The sensitivity of diffuse septal subendocardial LGE in diagnosing cardiac AL-amyloidosis was 88% with a specificity of 99% vs. HCM and 100% vs. HT. The sensitivity and specificity of myocardial nulling prior to blood pool was 77% and 100%, respectively, both vs. HCM and vs. HT. Maximal wall thickness with a cut-off of 14 mm had high diagnostic accuracy when differentiating cardiac AL-amyloidosis from HT: sensitivity 94%, specificity 89%. The sensitivities of reduced global longitudinal strain plus apical sparing were low with high specificities. CMR-based strain analysis was applicable for detecting apical sparing in cardiac AL-amyloidosis patients.”

3) The results on the diagnostic performance of LV basal segmental strain are additional or secondary findings given the high variability and low reproducibility of basal strain values in the LV. Hence, the manuscript should not be focused or centered on these findings and these findings should not be shown in the abstract or conclusion section.

Thank you for your advice. We have deleted the results regarding basal strain values from the abstract. Our results of interobserver analyses show that intraclass correlation of basal LS was good, near to excellent (0.87), the intraclass correlation of basal CS was excellent (0.92) (S1 Table). Thus we believe that the results regarding basal strain values are relevant enough to be included in the manuscript beside the main findings.

4) In line with the above-mentioned comments, please re-edit the title of the study as “Potential clinical relevance of cardiac magnetic resonance to diagnose cardiac light chain amyloidosis”.

Thank you for your suggestion, we have changed the title as requested.

---

## [Editor Report · Decision Letter 4]

3 May 2022

PONE-D-21-24457R4Potential clinical relevance of cardiac magnetic resonance to diagnose cardiac light chain amyloidosis.PLOS ONE

Dear Dr. Vago,

Thank you for submitting your manuscript to PLOS ONE. After careful consideration, we feel that it has merit but does not fully meet PLOS ONE’s publication criteria as it currently stands. Therefore, we invite you to submit a revised version of the manuscript that addresses the points raised during the review process.

Please include the following items when submitting your revised manuscript:A rebuttal letter that responds to each point raised by the academic editor and reviewer(s). You should upload this letter as a separate file labeled 'Response to Reviewers'.A marked-up copy of your manuscript that highlights changes made to the original version. You should upload this as a separate file labeled 'Revised Manuscript with Track Changes'.An unmarked version of your revised paper without tracked changes. You should upload this as a separate file labeled 'Manuscript'.We look forward to receiving your revised manuscript.

Kind regards,

Daniel A. Morris, M.D

Academic Editor

PLOS ONE

Additional Editor Comments:

1) Thank you very much for your efforts to improve the manuscript, which in fact has significantly improved. However, it remains the same issues as in the previous submissions. In this respect, the findings on the role of basal LS and CS to differentiate AL-Amyloidosis from CMH and HT are not clinically relevant. In this respect, a cutoff of basal LS at 21 and 16% and basal CS at 31% are within the range of normality, and thus, these cutoffs have low specificity (i.e., in other words, a high proportion of healthy subjects have values of basal LS about 16% and 21% and basal CS about 31% or even lower). In addition, the variability day to day of basal LS and basal CS is well known in clinical practice. Hence, as it has been stated and highlighted in previous submissions, the findings of the diagnostic performance of basal LS and CS or any strain parameter are of low clinical relevance to differentiating Al-Amyloidosis from CMH or HT and thus, these findings should play a secondary role in the manuscript.

2) Taking into consideration the above-mentioned issues please re-edit and refine the abstract, conclusions, and the first paragraph of the discussion section.

3) Please consider using an abstract like this:

- Background: While patients with cardiac transthyretin amyloidosis are easily diagnosed with bone scintigraphy, the detection of cardiac light chain (AL) amyloidosis is challenging. Cardiac magnetic resonance (CMR) analyses play an essential role in the differential diagnosis of cardiomyopathies; however, limited data are available from cardiac AL-Amyloidosis. Hence, the purpose of the present study was to analyze the potential role of CMR in the detection of cardiac AL-amyloidosis.

Methods: We included 35 patients with proved cardiac AL-amyloidosis and two control groups constituted by 330 patients with hypertrophic cardiomyopathy (HCM) and 70 patients with arterial hypertension (HT), who underwent CMR examination. The phenotype and degree of left ventricular (LV) hypertrophy and the amount and pattern of late gadolinium enhancement (LGE) were evaluated. In addition, global and regional LV strain parameters were also analyzed using feature-tracking techniques. Sensitivity and specificity of several CMR parameters were analyzed in diagnosing cardiac AL-amyloidosis.

Results: The sensitivity and specificity of diffuse septal subendocardial LGE in diagnosing cardiac AL-amyloidosis was 88% and 100%, respectively. Likewise, the sensitivity and specificity of septal myocardial nulling prior to blood pool was 71% and 100%, respectively. In addition, a maximal LV end-diastolic septal wall thickness ≥ 14 mm had an optimal diagnostic performance to differentiate cardiac AL-amyloidosis from HT (sensitivity 91%, specificity 83%). On the other hand, a reduced global LV longitudinal strain (< 15%) plus apical sparing had a very low sensitivity (6%) in detecting AL-Amyloidosis, but with very high specificity (100%).

Conclusions: The findings from this study suggest that CMR could have an optimal diagnostic performance in the diagnosis of cardiac AL-amyloidosis. Hence, further larger studies are warranted to validate the findings from this study.

4) Please consider using a conclusion like this:

- The findings from this study suggest that CMR could have an optimal diagnostic performance in the diagnosis of cardiac AL-amyloidosis. In this respect, the sensitivity and specificity of diffuse septal subendocardial LGE in diagnosing cardiac AL-amyloidosis was 88% and 100% and of septal myocardial nulling prior to blood pool was 71% and 100%, respectively. In addition, a maximal LV end-diastolic septal wall thickness ≥ 14 mm had an optimal diagnostic performance to differentiate cardiac AL-amyloidosis from HT (sensitivity 91%, specificity 83%). On the other hand, a reduced global LV longitudinal strain (< 15%) plus apical sparing had a very low sensitivity (6%) in detecting AL-Amyloidosis, but with very high specificity (100%). Hence, further larger studies are warranted to validate the potential key role of CMR in the diagnosis of cardiac AL-amyloidosis.

5) Please consider using in the first sentence and paragraph of the discussion section the following statements:

- Analyzing a cohort of 35 patients with proved cardiac AL-amyloidosis and two control groups constituted by 330 patients with hypertrophic cardiomyopathy (HCM) and 70 patients with arterial hypertension (HT), who underwent CMR examination, the findings from this study suggest that CMR could have an optimal diagnostic performance in the diagnosis of cardiac AL-amyloidosis. In this respect, the sensitivity and specificity of diffuse septal subendocardial LGE and of septal myocardial nulling prior to blood pool in diagnosing cardiac AL-amyloidosis was excellent. In addition, a maximal LV end-diastolic septal wall thickness ≥ 14 mm had an optimal diagnostic performance to differentiate cardiac AL-amyloidosis from HT. On the other hand, a reduced global LV longitudinal strain (< 15%) plus apical sparing had a very low sensitivity (6%) in detecting AL-Amyloidosis, but with very high specificity (100%).

---

## [Author Response · Author response to Decision Letter 4]

16 May 2022

We thank the editor for his thorough work and detailed suggestions for improving the manuscript. We have changed the abstract, the conclusion, and the first sentence and paragraph of the discussion section as requested. We only deviated from the editor's proposal on one point: Data show that a LV end-diastolic septal wall thickness ≥ 15 mm had better diagnostic performance to differentiate cardiac AL-amyloidosis from HT than a cut-off of 14 mm (15 mm: sensitivity 91%, specificity 89%, AUC 0.900, 14 mm: sensitivity 91%, specificity 83%, AUC 0.870). However, if maximal end-diastolic wall thickness is analysed (not necessarily septal), the optimal cut-off is 14 mm (sensitivity 94%, specificity 89%, AUC 0.927). Since the maximal end-diastolic wall thickness ≥ 14 mm had the highest diagnostic performance of the above mentioned parameters, we decided to highlight it. Max. EDWT has also been added to tables 2 and 3.

---

## [Editor Report · Decision Letter 5]

23 May 2022

PONE-D-21-24457R5Potential clinical relevance of cardiac magnetic resonance to diagnose cardiac light chain amyloidosis.PLOS ONE

Dear Dr. Vago,

Thank you for submitting your manuscript to PLOS ONE. After careful consideration, we feel that it has merit but does not fully meet PLOS ONE’s publication criteria as it currently stands. Therefore, we invite you to submit a revised version of the manuscript that addresses the points raised during the review process.

Please include the following items when submitting your revised manuscript:A rebuttal letter that responds to each point raised by the academic editor and reviewer(s). You should upload this letter as a separate file labeled 'Response to Reviewers'.A marked-up copy of your manuscript that highlights changes made to the original version. You should upload this as a separate file labeled 'Revised Manuscript with Track Changes'.An unmarked version of your revised paper without tracked changes. You should upload this as a separate file labeled 'Manuscript'.We look forward to receiving your revised manuscript.

Kind regards,

Daniel A. Morris, M.D

Academic Editor

PLOS ONE

Journal Requirements:

Additional Editor Comments:

I would like to congratulate again to the authors for the effort made to improve this interesting study. In fact, the study is of high interest for the Journal and it will be for all medical community, given the originality the findings regarding the potential usefulness of CMR in the diagnosis of cardiac Al-Amyloidosis.

Just one last minor correction is pending. In this respect, please delete in the text, abstract, conclusions, and tables the “new” analysis titled “maximal LV end-diastolic wall thickness” since in clinical practice almost never are measured all walls thickness but only the septal and posterior wall thickness, which are easy correlated with the measurements from echocardiography. By the way, as the authors have been stated, please change / correct in the abstract, conclusion, and first paragraph of the discussion section the sentence “In addition, a maximal LV end-diastolic wall thickness ≥ 14 mm had an optimal diagnostic performance to differentiate cardiac AL-amyloidosis from HT (sensitivity 94%, specificity 89%)” by “In addition, a LV end-diastolic septal wall thickness ≥ 15 mm had an optimal diagnostic performance to differentiate cardiac AL-amyloidosis from HT (sensitivity 91%, specificity 89%)”.

---

## [Author Response · Author response to Decision Letter 5]

25 May 2022

Journal Requirements:

Authors’ answer:

We have reviewed the reference list and made no changes.

Additional Editor Comments:

I would like to congratulate again to the authors for the effort made to improve this interesting study. In fact, the study is of high interest for the Journal and it will be for all medical community, given the originality the findings regarding the potential usefulness of CMR in the diagnosis of cardiac Al-Amyloidosis.

Just one last minor correction is pending. In this respect, please delete in the text, abstract, conclusions, and tables the “new” analysis titled “maximal LV end-diastolic wall thickness” since in clinical practice almost never are measured all walls thickness but only the septal and posterior wall thickness, which are easy correlated with the measurements from echocardiography. By the way, as the authors have been stated, please change / correct in the abstract, conclusion, and first paragraph of the discussion section the sentence “In addition, a maximal LV end-diastolic wall thickness ≥ 14 mm had an optimal diagnostic performance to differentiate cardiac AL-amyloidosis from HT (sensitivity 94%, specificity 89%)” by “In addition, a LV end-diastolic septal wall thickness ≥ 15 mm had an optimal diagnostic performance to differentiate cardiac AL-amyloidosis from HT (sensitivity 91%, specificity 89%)”.

Authors’ answer:

We thank again the editor for his thorough work and detailed suggestions for improving the manuscript. We have changed the abstract, the conclusion, and the first sentence and paragraph of the discussion section as requested. We have deleted max. EDWT in tables 2 and 3.

---

## [Editor Report · Decision Letter 6]

30 May 2022

Potential clinical relevance of cardiac magnetic resonance to diagnose cardiac light chain amyloidosis.

PONE-D-21-24457R6

Dear Dr. Vago,

We’re pleased to inform you that your manuscript has been judged scientifically suitable for publication and will be formally accepted for publication once it meets all outstanding technical requirements.

Kind regards,

Daniel A. Morris, M.D

Academic Editor

PLOS ONE

---

## [Editor Report · Acceptance letter]

3 Jun 2022

PONE-D-21-24457R6 

Potential clinical relevance of cardiac magnetic resonance to diagnose cardiac light chain amyloidosis. 

Dear Dr. Vago:

I'm pleased to inform you that your manuscript has been deemed suitable for publication in PLOS ONE. Congratulations! Your manuscript is now with our production department. 

Kind regards, 

on behalf of

Dr. Daniel A. Morris 

Academic Editor

PLOS ONE